# Quantifying the noise sensitivity of the Wasserstein metric for images

Erik Lager [1]  Gilles Mordant [2]  Amit Moscovich [1]

## Abstract

Wasserstein metrics are increasingly adopted as similarity scores for images. We consider the sensitivity of Wasserstein metrics with respect to pixel-wise additive noise when the images are treated as discrete measures on the pixel grid. We derive finite-sample expectation bounds for a Gaussian noise model. Among other results, we prove that the error in the signed 2-Wasserstein discrepancy scales with the square root of the noise standard deviation. This is favorable compared to the Euclidean metric that scales linearly, and thus provides a theoretical basis for the benefits of optimal transport distances in noisy settings. We present experiments that support our theoretical findings and point to a peculiar phenomenon where increasing the level of noise can decrease the Wasserstein distance. A case study on cryo-electron microscopy images demonstrates that the Wasserstein metric can capture the geometry of the data manifold in high noise settings even when the Euclidean metric fails.

## 1. Introduction

Optimal Transport (OT) provides a principled way to measure the distance between probability measures, capturing not only pointwise differences but also the underlying geometry of the data. Recent advances in computational approximation methods (Cuturi, 2013; Schmitzer, 2019) contributed greatly to the rising popularity of optimal transport across many domains, such as computer vision (Feydy et al., 2019), domain adaptation (Courty et al., 2017), and others. In imaging applications, the Wasserstein metric can be used to measure similarity by treating images as discrete measures on a grid, assigning a point mass to every pixel, proportional to its value. One field where this approach is gaining popularity is in single-particle cryo-electron microscopy (cryo-EM), a domain characterized by extremely high noise levels, where OT-based methods have been successfully applied to fundamental tasks, including the alignment of 3D density maps (Riahi et al., 2023; Singer & Yang, 2024), the clustering of 2D tomographic projections (Rao et al., 2020), and the rotational alignment of tomographic projections with heterogeneity (Shi et al., 2025). We believe that a driver for this adoption is that, empirically, the Wasserstein metric appears more robust to noise than the standard Euclidean norm.

**Related work.** In generative modeling, OT-based metrics have inspired methods such as Wasserstein GAN (Arjovsky et al., 2017), Wasserstein autoencoders (Tolstikhin et al., 2019) and flow matching (Lipman et al., 2023; Albergo & Vanden-Eijnden, 2023; Liu et al., 2023). The latter in particular has strong connections to OT in its dynamic formulation. Building upon this, recent variants of flow matching incorporate OT solvers into the training process (Tong et al., 2024; Chemseddine et al., 2025; Zhang et al., 2025; Mousavi-Hosseini et al., 2026). While our work does not target these models specifically, we believe that a better understanding of the noise robustness of optimal transport procedures is needed to understand why modern generative models work so well. We discuss alternative models to ours in the last paragraph of Section 2.

Many variants of OT such as partial optimal transport (Chapel et al., 2020; Raghvendra et al., 2024) and unbalanced optimal transport (Benamou et al., 2015; Chizat et al., 2018) have been proposed to address mass imbalance. Our work can be extended in that direction, see Appendix B.

**Our contribution.** To the best of our knowledge, this is the first paper that studies the noise robustness of Wasserstein metrics for measures on a fixed grid[1]. On the theoretical side, we provide quantitative bounds relating the signed Wasserstein cost (see equation (4)) between noise-corrupted

---

[1]Department of Statistics and Operations Research, Tel Aviv University, Tel Aviv, Israel [2]Applied and Computational Mathematics, Yale University, New Haven, USA. Correspondence to: Erik Lager <erikice1@gmail.com>, Gilles Mordant <gilles.mordant@yale.edu>, Amit Moscovich <mosco@tauex.tau.ac.il>.

*Proceedings of the 43$^{rd}$ International Conference on Machine Learning*, Seoul, South Korea. PMLR 306, 2026. Copyright 2026 by the author(s).

[1]Following a request by an anonymous referee, we stress that the problem of interest in this paper differs from the setting of two point clouds where the location of the points is corrupted by additive noise.

images and the signed Wasserstein cost between the clean images. Focusing on a Gaussian noise model with fixed mass and pixel-wise standard deviation proportional to $\sigma$, we show that the signed $p$-Wasserstein discrepancy between a noise-corrupted $n \times n$ picture and its clean counterpart gives rise to an error term that scales like $(n\sigma)^{1/p}$, see Theorem 3.5. For the 1-Wasserstein distance, considering a similar noise model, Theorem 3.7 establishes that the distance between two noisy pictures deviates by at most order $\sigma n \log_2 n$ from the distance between the clean ones. Theorem 3.8 gives a bound for the case of two different measures and $p \geq 1$. We complement our theoretical results with simulations in Section 4, showcasing the properties of the signed Wasserstein discrepancy in a variety of cases.

## 2. Wasserstein metrics for signed and noise corrupted measures

**Wasserstein metric.** Consider two probability measures $\mu, \nu \in \mathcal{P}(\mathcal{X})$. For any $1 \leq p \in \mathbb{N}$ and given a ground cost $\mathsf{d} : \mathcal{X} \times \mathcal{X} \to \mathbb{R}_+$, the Wasserstein metric between $\mu$ and $\nu$ is defined as

$$W_p(\mu, \nu) := \left( \inf_{\pi \in \Gamma(\mu,\nu)} \int_{\mathcal{X} \times \mathcal{X}} \mathsf{d}(x,y)^p \mathrm{d}\pi(x,y) \right)^{\frac{1}{p}} \quad (1)$$

where $\Gamma(\mu, \nu)$ is the set of measures with respective marginals $\mu$ and $\nu$. This definition extends to non-probability measures as long as they have equal total mass.

Under mild conditions, the Wasserstein metric admits a dual formulation, (Santambrogio, 2015, Theorem 1.39)

$$W_p^p(\mu, \nu) = \sup_{f \in L^1(\mu)} \int_{\mathcal{X}} f(x)\mathrm{d}\mu(x) + \int_{\mathcal{X}} f^{\mathsf{d}_p}(y)\mathrm{d}\nu(y),$$
$$(2)$$

where $f^{\mathsf{d}_p}(y) := \inf_{x \in \mathcal{X}} \left( \mathsf{d}(x,y)^p - f(x) \right)$.

In the case of the 1-Wasserstein metric, the dual formulation further admits the simplified form

$$W_1(\mu, \nu) = \sup_{f \in \mathrm{Lip}_1(\mathcal{X})} \langle f, \mu - \nu \rangle, \quad (3)$$

where $\mathrm{Lip}_1(\mathcal{X})$ is the set of 1-Lipschitz functions with respect to $\mathsf{d}$ on $\mathcal{X}$. The dual formulations are particularly useful to study stability of optimal transport with respect to perturbations of the marginals $\mu$ and $\nu$.

**Extension to signed measures.** In this section we explain how the definition of Wasserstein metrics can be extended to signed measures. This is necessary for two reasons: first, some image modalities (such as cryo-EM) naturally involve negative pixels. Second, even when all pixels are non-negative, once noise is introduced, e.g. pixel-wise i.i.d.

Gaussian noise, negative values may appear. Since we identify pixel values with point masses, this means we must account for signed measures.

Let $\mu, \nu$ be two signed measures with Jordan decompositions $\mu = \mu_+ - \mu_-$ and $\nu = \nu_+ - \nu_-$. Mainini (2012) considered the two (positive) measures,

$$S_{\mu,\nu} = \mu_+ + \nu_- \quad \text{and} \quad T_{\mu,\nu} = \nu_+ + \mu_-, \quad (4)$$

and defined a *signed Wasserstein discrepancy* between $\mu, \nu$,

$$W_p^{\pm}(\mu, \nu) := W_p(S_{\mu,\nu}, T_{\mu,\nu}). \quad (5)$$

Note that if the masses of the signed measures $\mu$ and $\nu$ are equal, then the positive measures $S_{\mu,\nu}, T_{\mu,\nu}$ have equal masses. Thus, $W_p(S_{\mu,\nu}, T_{\mu,\nu})$ is well defined. The signed Wasserstein discrepancy $W_p^{\pm}$ is a metric when $p = 1$ but not when $p > 1$, since it does not satisfy the triangle inequality (Mainini, 2012, Proposition 3.4). See Figure 5 for a surprising consequence of this.

For other applications and variants of signed Wasserstein discrepancies, see Groppe et al. (2025); Engquist et al. (2016); Thorpe et al. (2017).

**The issue of noise.** We model images as real-valued signals on a square grid $G_n$ of $n^2$ pixels which we identify with signed discrete measures. The aim of this work is to investigate how $W_p^{\pm}$ behaves when the images/measures $\mu$ and $\nu$ are corrupted by noise. In particular, consider observing

$$\mu_\varepsilon := \mu + \varepsilon_\mu \quad \text{and} \quad \nu_\varepsilon := \nu + \varepsilon_\nu, \quad (6)$$

and let

$$S_{\mu_\varepsilon,\nu_\varepsilon} := (\mu_\varepsilon)_+ + (\nu_\varepsilon)_- \quad (7)$$
$$T_{\mu_\varepsilon,\nu_\varepsilon} := (\nu_\varepsilon)_+ + (\mu_\varepsilon)_- . \quad (8)$$

Further set

$$C_S := \sum_{x \in G_n} \left( (\mu_\varepsilon)_+(x) + (\nu_\varepsilon)_-(x) \right) \quad (9)$$

$$C_T := \sum_{x \in G_n} \left( (\nu_\varepsilon)_+(x) + (\mu_\varepsilon)_-(x) \right). \quad (10)$$

Normalizing $S_{\mu_\varepsilon,\nu_\varepsilon}$ by $C_S$ and $T_{\mu_\varepsilon,\nu_\varepsilon}$ by $C_T$ is necessary to ensure that both measures have the same (unit) mass in the case where $\sum_{x \in G_n} \mu_\varepsilon(x) \neq \sum_{x \in G_n} \nu_\varepsilon(x)$. In the sequel, we will use the notation

$$\bar{S}_{\mu_\varepsilon,\nu_\varepsilon} := \frac{S_{\mu_\varepsilon,\nu_\varepsilon}}{C_S}, \qquad \bar{T}_{\mu_\varepsilon,\nu_\varepsilon} := \frac{T_{\mu_\varepsilon,\nu_\varepsilon}}{C_T}. \quad (11)$$

We aim at understanding the relationship between $W_p^{\pm}(\mu, \nu)$ and $W_p(\bar{S}_{\mu_\varepsilon,\nu_\varepsilon}, \bar{T}_{\mu_\varepsilon,\nu_\varepsilon})$. To put our analysis into context,

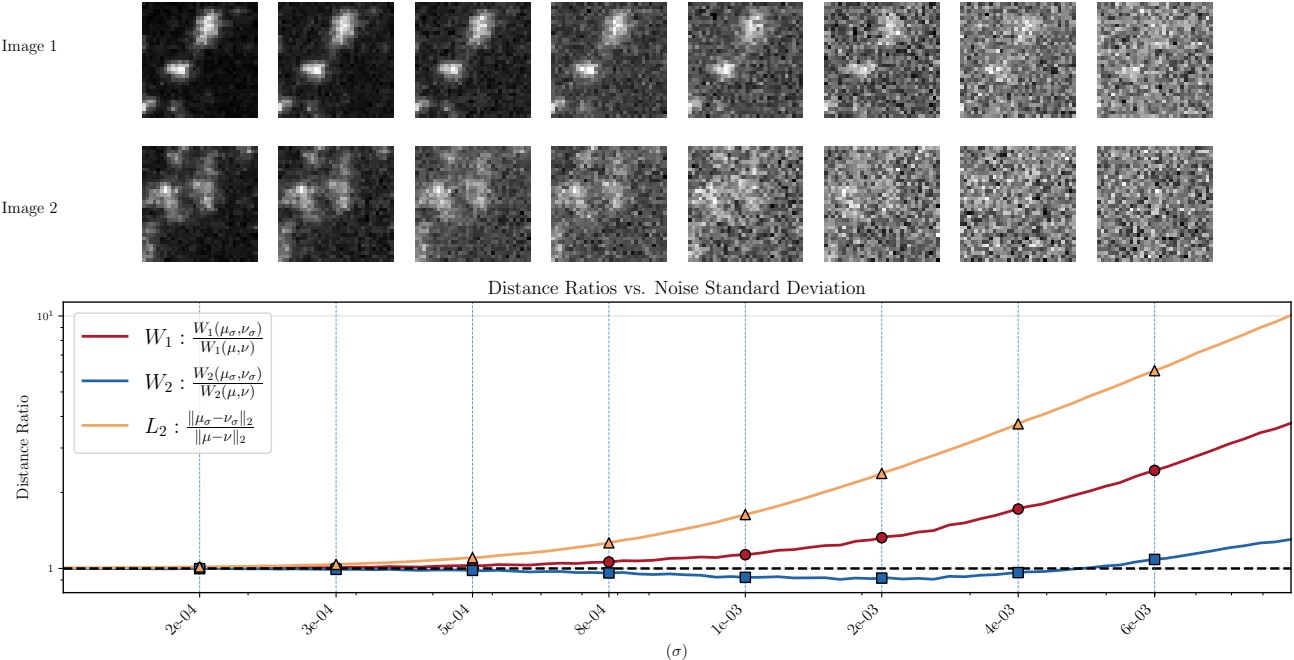

*Figure 1.* Distance ratios of $L^2$, $W_1$ and $W_2$ on a pair of noisy images as a function of the noise level. $L^2$ diverges first, followed by $W_1$ and lastly, $W_2$ departs from the original distance between the images, exhibiting more noise robustness. Above each marker we show the pair of images that were compared using all 3 metrics. See Section 4.2 for more details.

consider the standard squared $L^2$ distance, a common metric for image comparison. In the presence of additive Gaussian noise with variance $\sigma^2$, the expected squared $L^2$ distance between a signal and its noisy version has a simple, direct relationship: it is exactly $n^2\sigma^2$. This metric, however, is local and insensitive to the underlying geometric structure of the signal. In contrast, the (signed) Wasserstein cost is claimed to capture this geometry, but its behavior under noise is far more complex to characterize. This paper aims to bridge that gap by providing a theoretical and empirical analysis of its robustness.

**Dyadic bound on the Wasserstein distance.** To get sharp estimates on the Wasserstein distance, the following proposition is particularly useful. This is Proposition 1 of (Weed & Bach, 2019), but on a domain with an arbitrary diameter (their formulation assumed $\mathrm{diam}(S) = 1$). The bound is based on the construction of a coupling at various scales, managing the mass imbalance in subdomains. This construction yields sharp rates in a variety of cases.

**Proposition 2.1.** *Let $\{\mathcal{Q}^k\}_{1 \leq k \leq k^*}$ be a dyadic partition of a set $S$ with parameter $\delta < 1$. Then, for probability measures $\mu$ and $\nu$ supported on $S$,*

$$\frac{W_p^p(\mu,\nu)}{\mathrm{diam}(S)^p} \leq \delta^{pk^*} + \sum_{k=1}^{k^*} \delta^{p(k-1)} \sum_{Q_i^k \in \mathcal{Q}^k} |(\mu - \nu)(Q_i^k)|.$$

Recall that a dyadic partition of a set $S$ with parameter $\delta < 1$ is a sequence $\{\mathcal{Q}^k\}_{1 \leq k \leq k^*}$ possessing the following properties. First, the sets in $\mathcal{Q}^k$ form a partition of $S$. Further, if $Q \in \mathcal{Q}^k$, then $\mathrm{diam}(Q) \leq \delta^k$. Finally, if $Q^{k+1} \in \mathcal{Q}^{k+1}$ and $Q^k \in \mathcal{Q}^k$, then either $Q^{k+1} \subset Q^k$ or $Q^{k+1} \cap Q^k = \emptyset$.

## 3. Theoretical contributions

Our main theoretical results are upper bounds in expectation on the effect that the noise has on the signed Wasserstein cost between images. To avoid boundary effects and simplify some of our analyses, we consider the pixel grid to have cyclic boundary conditions, i.e., the left–right and top–bottom edges wrap. With this choice, each pixel has the same number of neighbors. While the Wasserstein metric naturally extends to non-probability measures, it still requires that both measures have the same mass, as described in the previous subsection. A standard i.i.d. noise model comes with the need of rescaling the pictures, which we study in the following section.

*Proofs of the following results are collected in Appendix A.*

### 3.1. The impact of rescaling.

An important fact is that the signed Wasserstein discrepancy, by construction, has an intricate non-linear behavior in terms of the noise when the mass of the latter is not fixed. By

duality, observe that

$$W_p^p(\bar{S}_{\mu_\varepsilon,\nu_\varepsilon}, \bar{T}_{\mu_\varepsilon,\nu_\varepsilon}) = \sup_f \langle f, \bar{S}_{\mu_\varepsilon,\nu_\varepsilon}\rangle + \langle f^{\mathsf{d}_p}, \bar{T}_{\mu_\varepsilon,\nu_\varepsilon}\rangle$$

$$= \sup_f \frac{\langle f, S_{\mu_\varepsilon,\nu_\varepsilon}\rangle + \langle f^{\mathsf{d}_p}, T_{\mu_\varepsilon,\nu_\varepsilon}\rangle}{\sum_{x\in G_n} S_{\mu_\varepsilon,\nu_\varepsilon}(x)} + \quad (12)$$

$$+ \left(\frac{1}{\sum_{x\in G_n} T_{\mu_\varepsilon,\nu_\varepsilon}(x)} - \frac{1}{\sum_{x\in G_n} S_{\mu_\varepsilon,\nu_\varepsilon}(x)}\right)\langle f^{\mathsf{d}_p}, T_{\mu_\varepsilon,\nu_\varepsilon}\rangle.$$

This decomposition shows that the optimal dual function must balance two objectives at the same time: the first one is the transport problem, and the second can be interpreted as a mass imbalance penalization. In the case of i.i.d. Gaussian noise, the result above can be refined to yield,

**Theorem 3.1.** *Consider two $n \times n$ images $\mu$ and $\nu$. Assume that $\varepsilon_\mu, \varepsilon_\nu$ are $\mathcal{N}(0_{n^2}, \sigma^2 I_{n^2})$. Recall the definition of $\bar{S}_{\mu_\varepsilon,\nu_\varepsilon}, \bar{T}_{\mu_\varepsilon,\nu_\varepsilon}$ in* (11)*. Then,*

$$W_1(\bar{S}_{\mu_\varepsilon,\nu_\varepsilon}, \bar{T}_{\mu_\varepsilon,\nu_\varepsilon}) = \frac{1}{\sum_{x\in G_n} S_{\mu_\varepsilon,\nu_\varepsilon}(x)} \quad (13)$$

$$\times \sup_{f\in\mathrm{Lip}_1} \langle f, S_{\mu_\varepsilon,\nu_\varepsilon} - T_{\mu_\varepsilon,\nu_\varepsilon}(1 + \mathrm{O}_p(1/n))\rangle.$$

Even though the above result does not seem symmetric, we establish in the proof that

$$\sum_{x\in G_n} S_{\mu_\varepsilon,\nu_\varepsilon}(x) - T_{\mu_\varepsilon,\nu_\varepsilon}(x) = \mathrm{O}_p(\sigma n), \quad (14)$$

from which we deduce that the apparent absence of symmetry is merely an artifact of the proof.

In general, one can hope that the ratio $\sigma/n$ is small, so that the result suggests that understanding the quantity $\sup_f \langle f, S_{\mu_\varepsilon,\nu_\varepsilon}\rangle + \langle f^{\mathsf{d}}, T_{\mu_\varepsilon,\nu_\varepsilon}\rangle$ under a suitable choice of noise is a first step to take towards completely characterizing the impact of the noise.

Without rescaling, a natural idea is to rely on unbalanced transport metrics between $\mu_+ + \nu_-$ and $\nu_+ + \mu_-$. In that case, we can derive similar bounds as those in the paper, which we defer to Appendix B. Using unbalanced transport metrics usually comes with the need to choose additional parameters. Further, a complete study of a pseudo-metric relying on decomposition into positive and negative parts followed by unbalanced OT hasn't been carried out before; there is no such theory like that of Mainini (2012).

### 3.2. Noise model

The previous section invites us to consider a noise model for which it is not necessary to rescale the measures. To this end, we will consider slightly correlated Gaussian noise where we identify each coordinate of the Gaussian noise vector with a point on the regular grid $G_n$.

**Assumption 3.2.** Consider an image modeled as an $n \times n$ grid of pixels and set $m = n^2$. Assume that the noise vector

$N = (N_1, \ldots, N_m)$ is drawn from a multivariate normal distribution $\mathcal{N}(0, \Sigma)$, where the covariance matrix $\Sigma$ is an $m \times m$ matrix defined as

$$\Sigma_{ij} = \begin{cases} \sigma^2 & \text{if } i = j \\ -\frac{\sigma^2}{m-1} & \text{if } i \neq j. \end{cases} \quad (15)$$

Note that this noise model is equivalent to drawing the pixels independently from $\mathcal{N}\left(0, \sigma^2 m/(m-1)\right)$, calculating their mean, and then subtracting the mean from every pixel.

**Proposition 3.3** (Noise model properties). *Under Assumption* 3.2*, the following holds.*

1. *The marginal pixel distribution is $N_i \sim \mathcal{N}(0, \sigma^2)$.*

2. *The sum of pixels is zero : $\sum_{i=1}^m N_i = 0$.*

This last property allows us to focus on the impact of the noise, while setting aside the questions pertaining to rescaling the measures whose behavior was captured in Theorem 3.1.

### 3.3. Multiscale $W_p$ bound on a single image

We shall begin by proving bounds in the particular case where we compare one image with a noise corrupted version of itself. We start with the case of $p = 1$.

**Proof sketch for upper bounds.** Our derivation relies on a multiscale argument using a dyadic partition of the pixel grid. We construct a suboptimal coupling by recursively matching mass imbalance across four quadrants at each scale. By summing the expected costs across all $\log_2 n$ levels of the partition, we obtain sharp bounds.

**Theorem 3.4.** *Let $\mu : G_n \to [0, 1]$ be a probability measure on the $n \times n$ unit grid $G_n$ with cyclic boundary conditions. Let $\varepsilon_1, \varepsilon_2$ be independent random signed measures on the grid that satisfy Assumption* 3.2 *and for convenience assume that $n = 2^\eta$. Then*

$$\frac{n\sigma}{\sqrt{\pi}} \leq \mathbb{E}W_1^\pm(\mu + \varepsilon_1, \mu + \varepsilon_2) \leq \frac{2\sqrt{2}\log_2 n + 1/\sqrt{2}}{\sqrt{\pi}} n\sigma. \quad (16)$$

**Proof sketch for $W_1$ lower bound.** We construct a potential function $f(x)$ that takes values $\pm\frac{1}{2n}$ depending on the sign of the noise at each pixel. This is a 1-Lipschitz function. Hence, we may use the Kantorovich-Rubinstein dual formulation to obtain a lower bound in expectation over the noise.

It is further possible to prove a result for $p > 1$. The rates differ substantially, as is clear from the following theorem.

**Theorem 3.5.** *Let $\mu : G_n \to [0,1]$ be a probability measure on the $n \times n$ unit grid $G_n$. Let $\varepsilon_1, \varepsilon_2$ be independent random signed measures on the grid that satisfy Assumption 3.2. For convenience, we again assume that $n = 2^\eta$, for $\eta \in \mathbb{N}$. Then, for $p > 1$ with $p \in \mathbb{N}$,*

$$\mathbb{E}\left[\left(W_p^\pm(\mu + \varepsilon_1, \mu + \varepsilon_2)\right)^p\right] \leq \frac{4\sqrt{2}}{\sqrt{\pi}} n\sigma. \qquad (17)$$

*Therefore, by Jensen's inequality, and the lower bound part of the proof:*

$$C_p n^{\frac{2}{p}-1} \sigma^{\frac{1}{p}} \leq \mathbb{E} W_p^\pm(\mu + \varepsilon_1, \mu + \varepsilon_2) \leq \left(\frac{4\sqrt{2}}{\sqrt{\pi}} n\sigma\right)^{\frac{1}{p}}, \qquad (18)$$

*where*

$$C_p := \frac{2^{\frac{1}{p}-1}}{\sqrt{\pi}} \Gamma\left(\frac{1}{2p} + \frac{1}{2}\right). \qquad (19)$$

*Remark* 3.6. In both theorems above, the upper bound can be improved by removing the factor $\sqrt{2}$ if only one image is corrupted by noise.

### 3.4. Multiscale $W_p$ bound on two noisy images

We now consider the practically relevant setting where two different images are each corrupted by independent noise. Throughout, we assume that the noise model follows Assumption 3.2 and assume that both images have unit mass. Our object of interest is thus

$$W_p^\pm(\mu + \varepsilon_\mu, \nu + \varepsilon_\nu). \qquad (20)$$

In the case $p = 1$, one obtains the following result.

**Theorem 3.7.** *Let $\mu, \nu : G_n \to [0,1]$ be two probability measures on the $n \times n$ unit grid $G_n$ with cyclic boundary conditions and let $\varepsilon_\mu, \varepsilon_\nu : G_n \to \mathbb{R}$ be signed noise measures that satisfy Assumption 3.2. For convenience we assume that $n = 2^\eta$, for $\eta \in \mathbb{N}$. Then*

$$\mathbb{E}\left[W_1^\pm(\mu + \varepsilon_\mu, \nu + \varepsilon_\nu) - W_1^\pm(\mu, \nu)\right] \qquad (21)$$
$$\leq \frac{4n \log_2 n + n}{\sqrt{\pi}} \sigma + \frac{\sqrt{2}}{n}.$$

The proofs of the three theorems above come from a multiscale upper bound on the Wasserstein distance that depends solely on the mass differences (here, the pixels intensities) and therefore enables to control the impact of the noise.

Even though the Wasserstein 2-discrepancy is often used in applications and has nice theoretical properties in the continuous setting, such as the Brenier–McCann theorem (Brenier, 1991), its signed counterpart does not enjoy the same metric properties as the signed Wasserstein 1-distance, as was already hinted at in the introduction.

This absence of a triangle inequality underlies the particular form of the following result.

**Theorem 3.8.** *Let $\mu, \nu : G_n \to [0,1]$ be two probability measures on the $n \times n$ unit grid $G_n$ with cyclic boundary conditions and let $\varepsilon_\mu, \varepsilon_\nu : G_n \to \mathbb{R}$ be signed noise measures that satisfy Assumption 3.2. For convenience we assume that $n = 2^\eta, p > 1$, for $\eta \in \mathbb{N}$. Then,*

$$\mathbb{E}\left[W_p^\pm(\mu + \varepsilon_\mu, \nu + \varepsilon_\nu)\right] \qquad (22)$$
$$\leq \left(\frac{\sqrt{2}}{2}\right)^{1-\frac{1}{p}} W_1(\mu, \nu)^{\frac{1}{p}} + \frac{\sqrt{2}}{2}\left(\frac{4}{\sqrt{\pi}} n \log_2 n + \frac{1}{\sqrt{\pi}} n\right)^{\frac{1}{p}} \sigma^{\frac{1}{p}}.$$

The proof of this theorem requires to first relate the signed Wasserstein $p$ discrepancy to a 1-Wasserstein distance between the uncorrupted measures by exploiting suboptimal couplings. This then enables to use a multiscale bound for the remaining part that mostly pertains to noise.

## 4. Numerical experiments and results

### 4.1. Quantitative validation of noise scaling

The first experiment we conduct aims to quantitatively measure how the distance between an image and its noisy counterpart scales when increasing noise variance. This allows for a direct comparison between the empirical behavior of each metric and the theoretical scaling laws derived in Theorem 3.5. The results are reported as Figure 2. All transport costs calculated are exact and were calculated using the POT Python package (Flamary et al., 2021)

To this end, we performed 100 independent trials, each time selecting a new, random 32x32 pixel image from the DOTMark 1.0 MicroscopyImages dataset (Schrieber et al., 2017). For each image $\mu$, we generated a noisy version $\mu + \varepsilon$ by adding zero-sum noise $\varepsilon$ satisfying Assumption 3.2, with variances ranging from $10^{-7}$ to 1. We computed the difference between the original image and the noisy one for $L^2$, $W_2$ and $W_1$ imposing cyclic boundary conditions (toroidal topology). This empirical result, where the $W_2$ cost scales with an exponent of approximately 0.5, suggests that the bound derived in Theorem 3.5 correctly captures the behavior of the signed 2-Wasserstein as a function of the noise variability $\sigma$.

### 4.2. Visualizing robustness of inter-image distances

We now investigate how well the different metrics preserve the original distance between two images when the latter are progressively corrupted by noise. For this experiment, we selected two distinct $32 \times 32$ pixel images from the DOTMark dataset and simultaneously corrupted them with different instances of zero-sum additive noise with a standard deviation ranging from $10^{-7}$ to $10^{-1}$. At each noise

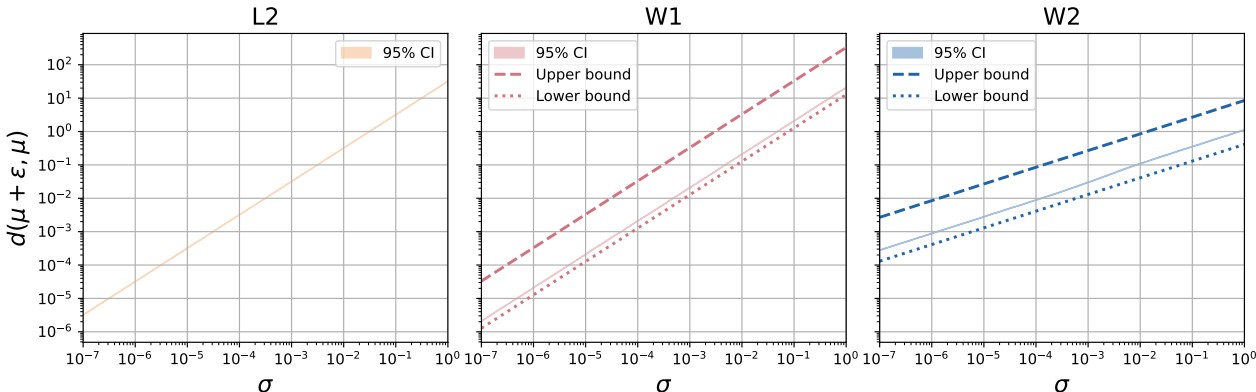

*Figure 2.* $L_2$, $W_1$ and $W_2$ discrepancies plotted against their theoretical upper and lower bounds as defined in Theorem 3.4 (for the $W_1$ bounds) and Theorem 3.5 (for the $W_2$ bound).

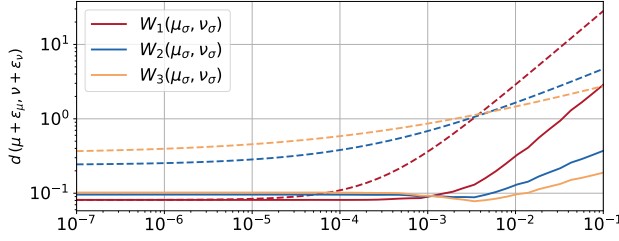

*Figure 3.* Distances between two randomly sampled images from the DOTMark microscopy dataset, both corrupted with noise sampled from the zero-sum normal distribution, in dashed (matching colors) we have the bounds for each $p$ from Theorem 3.5.

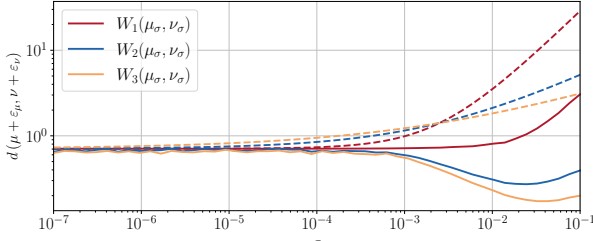

*Figure 4.* Distances between two noised images, one's mass concentrated at the pixel $(8, 8)$ and the other's at $(24, 24)$. In dashed matching colors we have the bounds for each $p$ from Theorem 3.8.

level, we computed the $W_1$, $W_2$ and $L^2$ discrepancies between the two noisy images. The results were averaged across 100 experiments. To evaluate stability, we computed a distance ratio by dividing the distance between the noisy images by the distance between the original, clean images. A ratio that remains close to 1 indicates robustness to noise. The output is displayed on Figure 1, which we already exhibited in the introduction. On that figure, the top panel visually depicts the degradation of the images as noise increases, while the main plot shows the distance ratio for each metric. The $L^2$ ratio (salmon-colored line) is the first to sharply diverge from 1, showing that the measured distance is quickly dominated by the noise. The $W_2$ ratio (blue line) is the most stable, remaining closest to the ideal ratio of 1 for the largest range of noise levels.

This experiment serves as a practical illustration of the scaling laws: as the $W_2$ discrepancy grows more slowly with noise, the underlying distance between the clean signals is better preserved.

**Visualisation of the bound of the inter-image distance.** To assess the bound established in Theorem 3.8, we have plotted the distance between two Microscopy images from

the DOTMark dataset being gradually corrupted by noise with the same parameter $\sigma$. We see in Figure 3 how tight the bound might be for $W_1$ (in the case of small noise) while it seems to not be tight for $W_2$ and $W_3$. We postulate that this is because the images used in this experiment are far from the "worst case scenario" in which the images are very similar to each other, or very far apart. We analyze these scenarios where the bound might be tighter in Figure 4.

**Characterizing metric behavior across image types.** While $W_2$ is robust, its behavior is not uniform. The purpose of the next experiment is to explore how the metrics' robustness varies across different classes of images and to highlight a key nuance of the signed $W_2$ metric. We repeated the distance ratio experiment from the previous section on four distinct image classes: white noise, typical cryo-EM projections, classic microscopy images, and synthetic images of two widely separated squares. The results are shown in Figure 5. In this figure, $W_2$ is shown to scale favorably. However, it also exhibits a lack of monotonicity with respect to the noise level. A phenomenon we study further in Section 4.4.

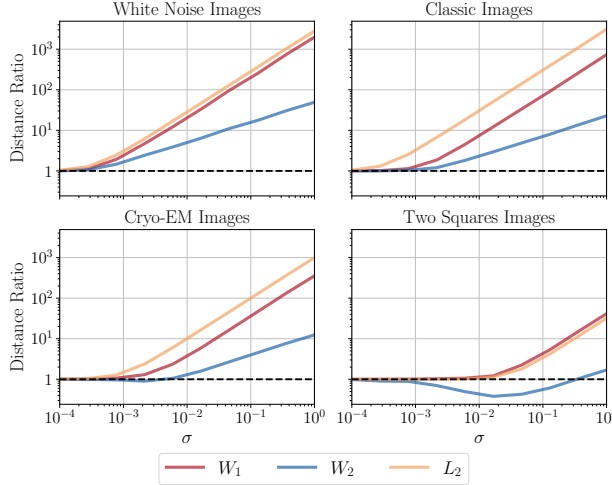

*Figure 5.* Ratios of the distance between the noisy images and the original images, for different kinds of images.

### 4.3. Analysis of cryo-electron microscopy images

Single-particle cryo-electron microscopy (cryo-EM) is a method for reconstructing the 3D structure of proteins and other large molecules. In this method, samples of a molecule of interest are frozen and then imaged using a transmission electron microscope. This results in many thousands of tomographic projections of the target molecule. The positions and orientations of the individual molecules are typically unknown and the images have extremely high levels of noise. Nonetheless, sophisticated computational methods were successful in recovering many different high-resolution 3D structures. Many important challenges remain. In particular, the reconstruction of flexible macromolecules with continuous degrees of freedom. See Bendory et al. (2020) for a survey of the computational challenges in cryo-EM.

In this section, we wish to demonstrate the potential benefit of Wasserstein metrics in the high-noise cryo-EM regime to the difficult task of recovering continuous conformational manifolds (Kileel et al., 2021). To this end, we generated 20 different projections of the E. Coli hsp90 protein (Shiau et al., 2006) in different conformational states using cryo-JAX (O'Brien et al., 2026). The location of the protein was shifted and the pictures were corrupted with high levels of noise to mimic the poor signal-to-noise ratio in real cryo electron microscopy images. For simplicity, all the images were normalized to sum to one. The goal is to assess how well each metric can recover known geometric relationships between particle images that undergo rotation and translation. To illustrate the difficulty of the task, Figure 6 shows a sample of the clean images and their noisy counterparts.

For each discrepancy, we compute all the pairwise distances, resulting in a 20x20 matrix which we can see in Figure 7. The top row shows the ground-truth distance matrices from the clean images, reflecting the structure of the transformations. The bottom row shows the matrices computed from their noisy counterparts. Under heavy noise, the $L^2$ distance matrix degrades into a random pattern, losing the original geometric structure. In contrast, the $W_2$ and $W_1$ cost matrices preserve the global diagonal structure of the ground-truth matrix.

**Benefits of $W_2$ over $W_1$** To evaluate the practical utility of using $W_2$ over $W_1$ in noisy settings, we design an experiment which contrasts the impact of structural changes with that of strong additive noise. While both discrepancies show better scaling compared to the $L_2$ metric, their relative distance rankings under pure additive noise like we see in Figure 7 are similar, masking the benefits of $W_2$.

In Figure 8, we compare a clean reference image to two variants: one that is spatially shifted (representing a genuine structural change, dotted) and one corrupted by pure Gaussian noise. We observe that $W_1$ is more easily overwhelmed by noise. In contrast, $W_2$ heavily penalizes the spatial shift due to its quadratic cost while "absorbing" local noise fluctuations. Our simulations indicate that $W_2$ can tolerate approximately 3.5x more noise than $W_1$ before it begins to confuse additive noise with a shift.

### 4.4. The decreasing distance phenomenon

Interestingly the estimated distance between images can even decrease when the noise increases. One can see an example in Figure 4 where for $p > 1$ we get a dip in the distance, showing that the images are getting closer together, similarly to the "two square images" in Figure 5. This phenomenon, which at first sight might be surprising, can be explained by the fact that for sparse pictures, the noise appearing between two structures can be used to "bridge" the transport distance between them, like we see in Figure 9. Instead of transporting mass across the entire distance between the two structures, the optimal transport plan utilizes the background noise as 'stepping stones.' Mass from structure A is moved to nearby noise peaks, while noise peaks near structure B are moved into structure B. This effectively shortens the total transport cost compared to the clean case.

## 5. Conclusion and future work

In this paper, we have investigated the behavior of the signed Wasserstein discrepancy under noise corruption of the pictures. Our theoretical contributions provide bounds for various situations of interest. In particular, certain bounds establish a better noise robustness of the signed Wasserstein discrepancy than the ubiquitous $L^2$ metric. Our numerical experiments on the DOTMark dataset corroborate these findings, with empirical results confirming that the $W_2$ discrepancy is more resilient to noise than both $L^2$ and $W_1$

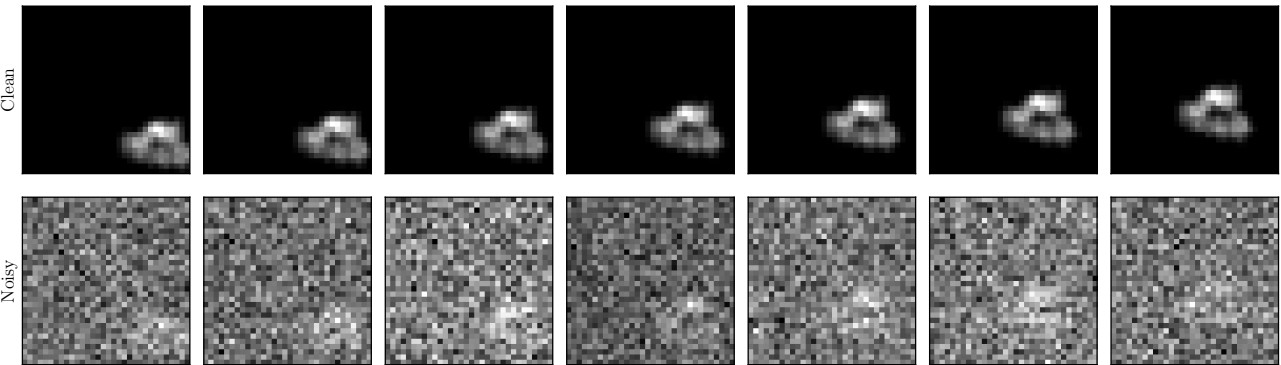

*Figure 6.* Projection images of the E. Coli Hsp90 molecule, with and without noise.

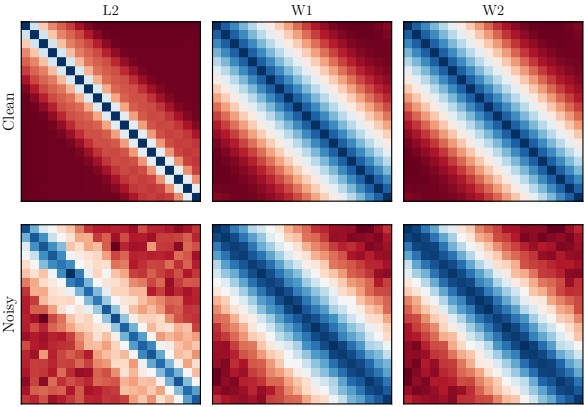

*Figure 7.* Top panel: Distance matrices between the different structures (noiseless). Bottom panel: Noisy mean distances over 100 experiments. Gradual blue-to-red gradients are better.

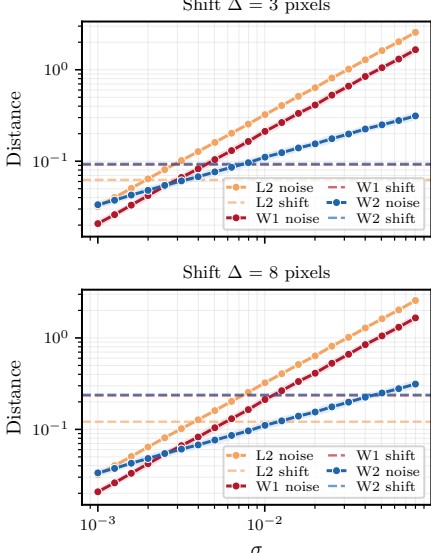

*Figure 8.* Sensitivity to additive noise versus spatial shifts. Calculated distances under pure Gaussian noise (solid lines) plotted against the noise standard deviation $\sigma$ across pixel shifts of $\Delta = 3$ (top) and $\Delta = 8$ (bottom).

distances. These results make a strong case for its use in noise-plagued applications like cryo-EM.

Despite these results, there remains room for additional work. A primary challenge would be to establish sharp bounds for $\mathbb{E}W_p^{\pm}(\mu_\varepsilon, \nu_\varepsilon) - \mathbb{E}W_p^{\pm}(\mu, \nu)$, which is hindered by the lack of triangle inequality. The numerical experiments further suggest that our bounds, despite capturing the correct behavior, are not tight. Finally, as our theory suggests that robustness increases with higher values of $p$, the interest of such choices of exponents for practical applications should be investigated in future works.

**Reproducibility.** The code for running the simulations and generating all the figures in this paper is available at: https://github.com/warik21/Quantifying-wasserstein-noise-sensitivity.

## Impact statement

It is our hope that this theoretical work will be a first step in understanding the robustness of optimal transport in high-noise imaging modalities such as cryo-electron microscopy or certain medical imaging applications, supporting methodological developments in these important fields.

We do not foresee any negative impacts. However, as a theoretical work, we admit that the wider societal impacts are pure speculation.

## Acknowledgments

AM is supported in part by ISF Grant No. 1662/22 and by NSF-BSF Grant No. 2022778.

We would like to thank Andrea Codegoni, Tobías I. Liaudat, Nir Sharon, Tal Wagner and the anonymous referees for interesting discussions and important feedback.

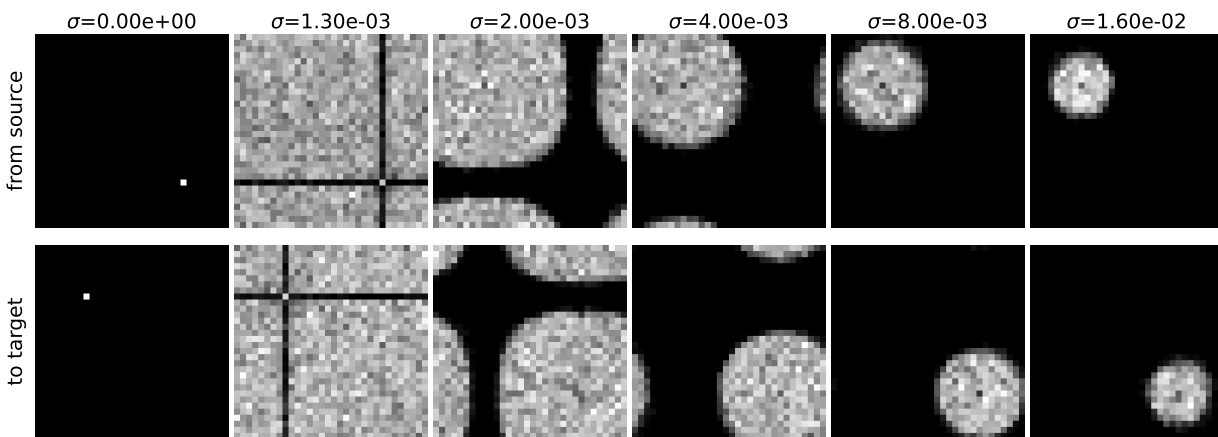

*Figure 9.* (Top panels) Where the mass of the original pixel (8,8) goes. (Bottom panels) Where the mass of the target pixel (24, 24) comes from in the optimal transport map between noisy versions of the single pixel images.

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

## A. Proofs

### A.1. Proof of Propositions

**Proposition 3.3** In the context of Assumption 3.2, the following holds.

1. The marginal distribution for each component is a Gaussian: $N_i \sim \mathcal{N}(0, \sigma^2)$.

2. The sum of the components is zero : $\sum_{i=1}^m N_i = 0$.

*Proof of Proposition 3.3.* We prove each point separately.

1. The marginal variance of each component $N_i$ is given by the diagonal entry $\Sigma_{ii}$, which is $\sigma^2$ by definition. Since the parent distribution is a multivariate normal with a mean vector of zero, each component is marginally distributed as $\mathcal{N}(0, \sigma^2)$.

2. We compute the variance of the sum of the components:

$$\text{Var}\left(\sum_{i=1}^m N_i\right) = \sum_{i,j} \text{Cov}(N_i, N_j) \tag{23}$$

$$= \sum_{i=1}^m \sum_{j=1}^m \Sigma_{ij} \tag{24}$$

$$= \sum_{i=1}^m \text{Var}(N_i) + \sum_{i \neq j} \text{Cov}(N_i, N_j) \tag{25}$$

$$= m \cdot \sigma^2 + m(m-1) \cdot \left(-\frac{\sigma^2}{m-1}\right) \tag{26}$$

$$= m\sigma^2 - m\sigma^2 = 0. \tag{27}$$

The expectation of the sum is

$$\mathbb{E}\left[\sum_{i=1}^m N_i\right] = \sum_{i=1}^m \mathbb{E}[N_i] = 0. \tag{28}$$

A random variable with zero mean and zero variance must be equal zero almost surely. Thus, $\sum_{i=1}^m N_i = 0$.

$\square$

**Proposition A.1** (Wasserstein Distance Decomposition). *Let $\mu$ and $\nu$ be two non-negative measures on a space $\mathcal{X}$ with equal total mass. It holds that*

$$W_p^p(\mu, \nu) \leq W_p^p\left((\mu - \nu)_+, (\nu - \mu)_+\right). \tag{29}$$

*Proof of Proposition A.1.* We can decompose any two measures $\mu$ and $\nu$ into a common part and two disjoint parts. Let $m$ be the largest measure such that for all Borel set $A$

$$m(A) \leq \mu(A) \text{ and } m(A) \leq \nu(A). \tag{30}$$

The remaining, disjoint parts of each measure are given by $\mu' := \mu - m = (\mu - \nu)_+$ as well as $\nu' := \nu - m = (\nu - \mu)_+$. Thus, we can write:

$$\mu = m + \mu' \qquad \nu = m + \nu' \tag{31}$$

Since $\mu$ and $\nu$ have the same total mass, it follows that $\mu'$ and $\nu'$ also have the same total mass.

We can then construct a valid transport plan $\pi$ from $\mu$ to $\nu$ by handling the common and disjoint parts separately. For the disjoint parts, let $\pi'_{\text{opt}}$ be the optimal transport plan from $\mu'$ to $\nu'$, whose cost is, by definition, $W_p^p(\mu', \nu')$. For the common part, we use the identity plan, $\pi_{\text{id}}$, which transports the mass at each point $x$ to itself. The cost of this plan is $\int_{\mathcal{X}} d(x, x)^p \mathrm{d}\pi_{\text{id}}(x) = 0$.

Using the gluing principle, we can form a complete transport plan $\pi = \pi_{\text{id}} + \pi'_{\text{opt}}$. This is a valid plan transporting $\mu$ to $\nu$. Its total cost is the sum of the costs of its components:

$$\text{Cost}(\pi) = \text{Cost}(\pi_{\text{id}}) + \text{Cost}(\pi'_{\text{opt}}) = 0 + W_p^p(\mu', \nu') \tag{32}$$

By the definition of the Wasserstein distance as the infimum of costs over all possible transport plans, the optimal cost must be less than or equal to the cost of this specific plan:

$$W_p^p(\mu, \nu) \leq W_p^p(\mu', \nu') \tag{33}$$

Substituting the definitions of $\mu'$ and $\nu'$ completes the proof.

$\square$

**Proposition A.2.** *Let $\mu : G_n \to [0, 1]$ be a probability measure on the $n \times n$ unit grid $G_n$ with cyclic boundary conditions and let $\varepsilon : G_n \to \mathbb{R}$ be a signed noise measure satisfying Assumption 3.2. Then, for $p > 1$,*

$$W_p^{\pm}(\mu, \mu + \varepsilon) \leq W_p(\varepsilon_-, \varepsilon_+), \tag{34}$$

*where $W_p$ denotes the standard extension of the Wasserstein distance to unnormalized non-negative measures of equal mass.*

*Proof of Proposition A.2.* By definition,

$$(W_p^{\pm})^p(\mu, \mu + \varepsilon) = W_p^p\big(\mu + (\mu + \varepsilon)_-, (\mu + \varepsilon)_+\big). \tag{35}$$

Thus, using Proposition A.1 on $\mu + (\mu + \varepsilon)_-$ and $(\mu + \varepsilon)_+$,

$$W_p^p\big(\mu + (\mu + \varepsilon)_-, (\mu + \varepsilon)_+\big) \tag{36}$$
$$\leq W_p^p\big((\mu + (\mu + \varepsilon)_- - (\mu + \varepsilon)_+)_+, \tag{37}$$
$$((\mu + \varepsilon)_+ - (\mu + (\mu + \varepsilon)_-))_+\big) \tag{38}$$
$$= W_p^p\big((\mu - ((\mu + \varepsilon)_+ - (\mu + \varepsilon)_-))_+, \tag{39}$$
$$((\mu + \varepsilon)_+ - (\mu + \varepsilon)_- - \mu))_+\big) \tag{40}$$
$$= W_p^p\big((\mu - (\mu + \varepsilon))_+, (\mu + \varepsilon - \mu)_+\big) \tag{41}$$
$$= W_p^p\big((-\varepsilon)_+, \varepsilon_+\big) = W_p^p(\varepsilon_-, \varepsilon_+). \qquad \square$$

**Proposition A.3.** *For any two images $\mu, \nu : G_n \to [0, \infty)$ and independent noises $\varepsilon_\mu, \varepsilon_\nu$ as in Assumption 3.2,*

$$W_1([\mu + \varepsilon_\mu - \nu - \varepsilon_\nu]_+, [\nu + \varepsilon_\nu - \mu - \varepsilon_\mu]_+) \tag{42}$$
$$\leq W_1(\mu, \nu) + W_1\big((\varepsilon_\mu - \varepsilon_\nu)_+, (\varepsilon_\mu - \varepsilon_\nu)_-\big).$$

*Proof of Proposition A.3.* By the Kantorovich–Rubinstein duality,

$$W_1\big([\mu + \varepsilon_\mu - \nu - \varepsilon_\nu]_+, [\nu + \varepsilon_\nu - \mu - \varepsilon_\mu]_+\big) = \tag{43}$$
$$= \sup_{\|f\|_{\mathrm{Lip}} \leq 1} \int f(\mu - \nu) + \int f(\varepsilon_\mu - \varepsilon_\nu).$$

For the first term, by KR duality,

$$\sup_{\|f\|_{\mathrm{Lip}} \leq 1} \int f(\mu - \nu) \leq W_1(\mu, \nu) \tag{44}$$

For the second term, via the Jordan decomposition,

$$\sup_{\|f\|_{\mathrm{Lip}} \leq 1} \int f(\varepsilon_\mu - \varepsilon_\nu) \leq W_1((\varepsilon_\mu - \varepsilon_\nu)_+, (\varepsilon_\mu - \varepsilon_\nu)_-). \tag{45}$$

Adding these together, we receive the desired bound. $\square$

**Proposition A.4.** *Let $\mu, \nu : G_n \to [0, \infty)$ be images on the square grid $G_n$ with spacing $h = 1/n$, and let $\varepsilon_\mu, \varepsilon_\nu$ satisfy Assumption 3.2. Identifying $G_n$ with the 2-torus, let $D := \mathrm{diam}(G_n) = \sqrt{2}/2$. Then, for any $p \geq 1$,*

$$W_p^\pm(\mu + \varepsilon_\mu, \nu + \varepsilon_\nu) \tag{46}$$
$$\leq D^{1-\frac{1}{p}}\big(W_1(\mu, \nu) + W_1(\varepsilon_+^*, \varepsilon_-^*)\big)^{\frac{1}{p}}, \quad \varepsilon^* := \varepsilon_\mu - \varepsilon_\nu$$

*Proof.* By definition of the signed distance,

$$W_p^\pm(\mu + \varepsilon_\mu, \nu + \varepsilon_\nu) = W_p\big((\mu + \varepsilon_\mu)_+ + (\nu + \varepsilon_\nu)_-, \tag{47}$$
$$(\nu + \varepsilon_\nu)_+ + (\mu + \varepsilon_\mu)_-\big).$$

Applying the decomposition inequality of Prop. A.1 (which "drops the overlap") to these nonnegative arguments gives

$$W_p^\pm(\mu + \varepsilon_\mu, \nu + \varepsilon_\nu) \leq W_p\big([\mu + \varepsilon_\mu - \nu - \varepsilon_\nu]_+, \tag{48}$$
$$[\nu + \varepsilon_\nu - \mu - \varepsilon_\mu]_+\big).$$

For general $p \geq 1$ on a bounded domain of diameter $D$ we use the standard comparison

$$W_p(\alpha, \beta) \leq D^{1-\frac{1}{p}} W_1(\alpha, \beta)^{\frac{1}{p}}. \tag{49}$$

Applying this to $(\alpha, \beta)$, where

$$\alpha = [\mu + \varepsilon_\mu - \nu - \varepsilon_\nu]_+, \tag{50}$$
$$\beta = [\mu + \varepsilon_\mu - \nu - \varepsilon_\nu]_-,$$

yields the following inequality:

$$W_p([\mu + \varepsilon_\mu - \nu - \varepsilon_\nu]_+, [\mu + \varepsilon_\mu - \nu - \varepsilon_\nu]_-) \tag{51}$$
$$\leq D^{1-\frac{1}{p}}\big(W_1([\mu + \varepsilon_\mu - \nu - \varepsilon_\nu]_+, [\mu + \varepsilon_\mu - \nu - \varepsilon_\nu]_-)\big)^{\frac{1}{p}}.$$

Using Proposition A.3, we conclude

$$D^{1-\frac{1}{p}}\big(W_1([\mu + \varepsilon_\mu - \nu - \varepsilon_\nu]_+, [\mu + \varepsilon_\mu - \nu - \varepsilon_\nu]_-)\big)^{\frac{1}{p}} \tag{52}$$
$$\leq D^{1-\frac{1}{p}}\big(W_1(\mu, \nu) + W_1((\varepsilon_\mu - \varepsilon_\nu)_+, (\varepsilon_\mu - \varepsilon_\nu)_-)\big)^{\frac{1}{p}}.$$

Since both $\varepsilon_\mu$ and $\varepsilon_\nu$ are normally distributed, we can say that $\varepsilon^* := \varepsilon_\mu - \varepsilon_\nu$ is also normally distributed, with $\mathrm{cov}(\varepsilon^*) = 2 \mathrm{cov}(\varepsilon_\mu)$. Thus,

$$D^{1-\frac{1}{p}}\big(W_1(\mu, \nu) + W_1((\varepsilon_\mu - \varepsilon_\nu)_+, (\varepsilon_\mu - \varepsilon_\nu)_-)\big)^{\frac{1}{p}}$$
$$\leq D^{1-\frac{1}{p}}\big(W_1(\mu, \nu) + W_1(\varepsilon_+^*, \varepsilon_-^*)\big)^{\frac{1}{p}}. \tag{53}$$
$$\square$$

### A.2. Proof of Theorems

**Theorem 3.1** Consider two $n \times n$ images $\mu$ and $\nu$. Assume that $\varepsilon_\mu, \varepsilon_\nu$ are $\mathcal{N}(0_{n^2}, \sigma^2 I_{n^2})$. Recall the definition of $\bar{S}_{\mu_\varepsilon, \nu_\varepsilon}, \bar{T}_{\mu_\varepsilon, \nu_\varepsilon}$ in (11). Then,

$$W_1(\bar{S}_{\mu_\varepsilon, \nu_\varepsilon}, \bar{T}_{\mu_\varepsilon, \nu_\varepsilon}) \tag{54}$$
$$= \frac{\sup_{f \in \mathrm{Lip}_1} \big\langle f, S_{\mu_\varepsilon, \nu_\varepsilon} - T_{\mu_\varepsilon, \nu_\varepsilon}\big(1 + O_p(\frac{1}{n})\big)\big\rangle}{\sum_{x \in G_n} S_{\mu_\varepsilon, \nu_\varepsilon}(x)}.$$

*Proof of Theorem 3.1.* First let us remark that

$$\sum_{x \in G_n} S_{\mu_\varepsilon, \nu_\varepsilon}(x) - T_{\mu_\varepsilon, \nu_\varepsilon}(x) \tag{55}$$
$$= \sum_{x \in G_n} \mu(x) + \varepsilon_\mu(x) - \nu(x) - \varepsilon_\nu(x)$$
$$= 0 + \sum_{x \in G_n} \varepsilon_\mu(x) - \varepsilon_\nu(x),$$

as

$$\sum_{x \in G_n} \mu_+(x) + \nu_-(x) = \sum_{x \in G_n} \nu_+(x) + \mu_-(x), \quad (56)$$

and thus

$$\sum_{x \in G_n} \mu(x) - \nu(x) = 0. \quad (57)$$

Remark that under our assumptions, for $N = n \cdot n$

$$\sum_{x \in G_n} \varepsilon_\mu(x) - \varepsilon_\nu(x) \sim \mathcal{N}(0, 2\sigma^2 N). \quad (58)$$

Because of this, one has that

$$\sum_{x \in G_n} S_{\mu_\varepsilon, \nu_\varepsilon}(x) = \sum_{x \in G_n} T_{\mu_\varepsilon, \nu_\varepsilon}(x) \left(1 + \frac{O_p(\sigma n)}{\sum_{x'} T_{\mu_\varepsilon, \nu_\varepsilon}(x')}\right). \quad (59)$$

Owing to our assumption on the signals, notice that

$$\sum_{x \in G_n} \mathbb{E} T_{\mu_\varepsilon, \nu_\varepsilon}(x) \geq n^2 \sigma / \sqrt{2\pi}. \quad (60)$$

Therefore,

$$W_1(\bar{S}_{\mu_\varepsilon, \nu_\varepsilon}, \bar{T}_{\mu_\varepsilon, \nu_\varepsilon}) = \sup_{f \in \text{Lip}_1} \langle \bar{S}_{\mu_\varepsilon, \nu_\varepsilon} - \bar{T}_{\mu_\varepsilon, \nu_\varepsilon}, f \rangle \quad (61)$$

$$= \sup_{f \in \text{Lip}_1} \left\langle \frac{S_{\mu_\varepsilon, \nu_\varepsilon}}{\sum_{x \in G_n} S_{\mu_\varepsilon, \nu_\varepsilon}(x)} - \frac{T_{\mu_\varepsilon, \nu_\varepsilon}}{\sum_{x \in G_n} T_{\mu_\varepsilon, \nu_\varepsilon}(x)}, f \right\rangle$$

$$= \frac{1}{\sum_x S_{\mu_\varepsilon, \nu_\varepsilon}(x)} \sup_{f \in \text{Lip}_1} \left\langle S_{\mu_\varepsilon, \nu_\varepsilon} \right.$$

$$\left. - T_{\mu_\varepsilon, \nu_\varepsilon} \left(1 + \frac{O_p(\sigma n)}{\sum_{x \in G_n} T_{\mu_\varepsilon, \nu_\varepsilon}(x)}\right), f \right\rangle.$$

$$\square$$

**Theorem 3.4** Let $\mu : G_n \to [0,1]$ be a probability measure on the $n \times n$ unit grid $G_n$ with cyclic boundary conditions. Let $\varepsilon_1, \varepsilon_2$ be independent random signed measures on the grid that satisfy Assumption 3.2. Then

$$\frac{n\sigma}{\sqrt{\pi}} \leq \mathbb{E} W_1^\pm(\mu + \varepsilon_1, \mu + \varepsilon_2) \leq \frac{2\sqrt{2} n \log_2 n}{\sqrt{\pi}} \sigma + \frac{n}{\sqrt{2\pi}} \sigma. \quad (62)$$

*Proof of Theorem 3.4.* By the Kantorovich–Rubinstein duality,

$$W_1^\pm(\mu, \mu + \varepsilon) = \sup_{f \in \text{Lip}_1} \langle f, \varepsilon \rangle = W_1(\varepsilon_+, \varepsilon_-). \quad (63)$$

$$W_1^\pm(\mu + \varepsilon_1, \mu + \varepsilon_2) = \sup_{f \in \text{Lip}_1} \langle f, \varepsilon_1 - \varepsilon_2 \rangle \quad (64)$$

$$= W_1((\varepsilon_1 - \varepsilon_2)_+, (\varepsilon_1 - \varepsilon_2)_-).$$

The first equality is the signed dual form with $\mu + \varepsilon_1 - (\mu + \varepsilon_2) = \varepsilon_1 - \varepsilon_2$. For simplicity, one can define $\varepsilon^* = \varepsilon_1 - \varepsilon_2$ such that $\mathbb{E}[|\varepsilon^*|] = \sqrt{2}\mathbb{E}[|\varepsilon_1|]$ as a sum of normally distributed random variables. Then, for the second equality, $\int \varepsilon^* = 0$ implies $\varepsilon^* = \varepsilon_+^* - \varepsilon_-^*$ with equal masses, so the balanced duality gives $W_1(\varepsilon_+^*, \varepsilon_-^*) = \sup_{f \in \text{Lip}_1} \langle f, \varepsilon^* \rangle$

**Proof of the upper bound.** Let $m = \varepsilon_+^*(G_n) = \varepsilon_-^*(G_n)$. By homogeneity of $W_1$,

$$W_1(\varepsilon_+^*, \varepsilon_-^*) = m W_1\left(\frac{\varepsilon_+^*}{m}, \frac{\varepsilon_-^*}{m}\right). \quad (65)$$

Apply Proposition 2.1 to the probability measures $\varepsilon_+/m$ and $\varepsilon_-/m$. There exists an integer $k^*$ with $k^* = \log_2 n$ such that

$$W_1\left(\frac{\varepsilon_+^*}{m}, \frac{\varepsilon_-^*}{m}\right) \leq \frac{\sqrt{2}}{2} 2^{-k^*} \quad (66)$$

$$+ \frac{\sqrt{2}}{2} \sum_{k=0}^{k^*} 2^{-(k-1)} \sum_{Q \in \mathcal{D}_k} \left|\left(\frac{\varepsilon_+^*}{m} - \frac{\varepsilon_-^*}{m}\right)(Q)\right|.$$

Multiplying by $m$ gives

$$W_1(\varepsilon_+^*, \varepsilon_-^*) \leq \frac{\sqrt{2}}{2} m \, 2^{-k^*} + \quad (67)$$

$$\frac{\sqrt{2}}{2} \sum_{k=0}^{k^*} 2^{-(k-1)} \sum_{Q \in \mathcal{D}_k} \left|\sum_{x \in Q} \varepsilon^*(x)\right|.$$

Taking expectations and using independence and zero mean of the noise,

$$\mathbb{E} W_1(\varepsilon_+^*, \varepsilon_-^*) \leq \frac{\sqrt{2}}{2} 2^{-k^*} \mathbb{E} m \quad (68)$$

$$+ \frac{\sqrt{2}}{2} \sum_{k=1}^{k^*} 2^{-(k-1)} \sum_{Q \in \mathcal{D}_k} \mathbb{E}\left|\sum_{x \in Q} \varepsilon^*(x)\right|.$$

Since each $\varepsilon^*(x)$ is Gaussian with variance $2\sigma^2$, one has $\mathbb{E}|\sum_{x \in Q} \varepsilon^*(x)| \leq \sqrt{2}\sigma\sqrt{|Q|}\sqrt{2/\pi}$ and $\mathbb{E}m = \sum_{x \in G_n} \mathbb{E}(\varepsilon^*(x))_+ = n^2 \sqrt{2}\sigma/\sqrt{2\pi}$. Furthermore, the dyadic family $\mathcal{D}_k$ has $|\mathcal{D}_k| = 2^{2k}$ cubes of cardinality $|Q| = n^2/2^{2k}$. Therefore

$$\sum_{Q \in \mathcal{D}_k} \mathbb{E}\left|\sum_{x \in Q} \varepsilon^*(x)\right| \leq \sqrt{2}\sigma\sqrt{\frac{2}{\pi}} \sum_{Q \in \mathcal{D}_k} \sqrt{|Q|} \quad (69)$$

$$= \sqrt{2}\sigma\sqrt{\frac{2}{\pi}} \cdot 2^{2k} \cdot \frac{n}{2^k} = 2\sigma\sqrt{\frac{1}{\pi}} n \, 2^k.$$

Plugging this into the multiscale sum yields

$$\frac{\sqrt{2}}{2} \sum_{k=0}^{k^*} 2^{-(k-1)} \sum_{Q \in \mathcal{D}_k} \mathbb{E}\left|\sum_{x \in Q} \varepsilon^*(x)\right| \leq \frac{\sqrt{2}}{2} 2\sigma\sqrt{\frac{1}{\pi}} n \sum_{k=1}^{k^*} 2$$

$$\leq 2\sqrt{2}\sigma\sqrt{\frac{1}{\pi}} n k^*. \quad (70)$$

With $k^* = \log_2 n$ this gives the $\sigma n \log_2 n$ contribution.

For the coarse term choose $k^*$ so that $2^{-k^*} = 1/n$. Then

$$\frac{\sqrt{2}}{2} 2^{-k^*} \mathbb{E}m = \frac{\sqrt{2}}{2} \frac{1}{n} \cdot \frac{n^2 \sqrt{2}\sigma}{\sqrt{2\pi}} = \frac{\sigma n}{\sqrt{2\pi}}, \qquad (71)$$

which is the $\sigma n$ contribution.

Collecting the two contributions and absorbing absolute constants into the displayed coefficients yields

$$\mathbb{E}W_1(\varepsilon_+^*, \varepsilon_-^*) \le \frac{2\sqrt{2}}{\sqrt{\pi}} n \log_2 n\sigma + \frac{1}{\sqrt{2\pi}} n\sigma. \qquad (72)$$

In this derivation the factor $m$ appears only in the coarse term and contributes to the $\sigma n$ piece after expectation. In the oscillation terms it cancels with the normalization, so no additional dependence on $m$ remains. There is no additive grid term independent of $\sigma$, hence no $1/(\sqrt{2}n)$ tail.

**Proof of the lower bound.** Let $f : G_n \to \mathbb{R}$ be the following,

$$f(x) := \begin{cases} -\frac{1}{2n} & \text{if } \varepsilon(x) < 0, \\ +\frac{1}{2n} & \text{if } \varepsilon(x) \ge 0. \end{cases} \qquad (73)$$

Since the distance between neighboring pixels is $1/n$ it follows that $f$ is 1-Lipschitz. Therefore, by the Kantorovich–Rubinstein duality,

$$W_1(\mu, \mu + \varepsilon) = W_1(\varepsilon_+, \varepsilon_-) \ge \langle f, \varepsilon_+ - \varepsilon_- \rangle \qquad (74)$$

Taking expectations on both sides and using the symmetry of $\varepsilon(x)$, we have

$$\mathbb{E}W_1(\mu, \mu + \varepsilon) \ge \mathbb{E}\langle f, \varepsilon_+ \rangle - \mathbb{E}\langle f, \varepsilon_- \rangle = 2\mathbb{E}\langle f, \varepsilon_+ \rangle. \qquad (75)$$

Recall that the marginal distribution $\varepsilon(x)$ is $\mathcal{N}(0, \sigma^2)$, and therefore conditioned on $\varepsilon_+(x) > 0$, we have $\mathbb{E}\varepsilon_+(x) = \sigma\sqrt{2/\pi}$ since that is the expectation of the half-normal distribution with variance $\sigma^2$. In expectation, $\langle f, \varepsilon_+ \rangle$ is a sum over $n^2/2$ pixels and its expectation satisfies

$$2\mathbb{E}\langle f, \varepsilon_+ \rangle = 2\mathbb{E}\left[ \sum_{x \text{ s.t. } \varepsilon(x)>0} f(x)\varepsilon_+(x) \right] \qquad (76)$$

$$= 2\frac{n^2}{2} \cdot \mathbb{E}\left[ f(x)\varepsilon_+(x) \mid \varepsilon_+(x) > 0 \right]$$

$$= n^2 \cdot \frac{1}{2n} \sqrt{\frac{2}{\pi}}\sigma = \frac{n\sigma}{\sqrt{2\pi}}.$$

Now, $W_1^\pm(\mu + \varepsilon_1, \mu + \varepsilon_2) = W_1^\pm(\mu, \mu + \varepsilon_2 - \varepsilon_1)$ but $\varepsilon_2 - \varepsilon_1$ is just a zero-mean noise vector that satisfies Assumption 3.2 but with double variance. It follows that

$$\mathbb{E}W_1^\pm(\mu + \varepsilon_1, \mu + \varepsilon_2) \ge \sqrt{2}\frac{n\sigma}{\sqrt{2\pi}} = \frac{n\sigma}{\sqrt{\pi}}. \qquad (77)$$

$\square$

**Theorem 3.5** Let $\mu : G_n \to [0, 1]$ be a probability measure on the $n \times n$ unit grid $G_n$. Let $\varepsilon_1, \varepsilon_2$ be independent random signed measures on the grid that satisfy Assumption 3.2. For convenience, we again assume that $n = 2^\eta$, for $\eta \in \mathbb{N}$. Then, for $p > 1$ with $p \in \mathbb{N}$,

$$\mathbb{E}\left[ \left(W_p^\pm(\mu + \varepsilon_1, \mu + \varepsilon_2)\right)^p \right] \le \frac{4\sqrt{2}}{\sqrt{\pi}} n\sigma. \qquad (78)$$

Therefore, by Jensen's inequality,

$$\mathbb{E}\left[ W_p^\pm(\mu + \varepsilon_1, \mu + \varepsilon_2) \right] \le \left( \frac{4\sqrt{2}}{\sqrt{\pi}} n\sigma \right)^{1/p}. \qquad (79)$$

*Proof of Theorem 3.5.* Building on Proposition A.2 and similarly to the proof of Theorem 3.4, we only need to upper bound $W_p(\varepsilon_+^*, \varepsilon_-^*)$ where $\varepsilon^* = \varepsilon_1 - \varepsilon_2$. By the assumption that the noise has total zero mass, this quantity is well defined.

Then, by the multiscale bound of Proposition 2.1

$$W_p^p(\varepsilon_+^*, \varepsilon_-^*) \qquad (80)$$

$$= \varepsilon_+^*(G_n)W_p^p\left( \frac{\varepsilon_+^*}{\varepsilon_+^*(G_n)}, \frac{\varepsilon_-^*}{\varepsilon_+^*(G_n)} \right)$$

$$\le 2^{-pk^* - p/2}\varepsilon_+^*(G_n)$$

$$+ 2^{-p/2} \sum_{k=1}^{k^*} 2^{-p(k-1)} \sum_{Q_i^k \in \mathcal{Q}^k} |\varepsilon_+^*(Q_i^k) - \varepsilon_-^*(Q_i^k)|$$

$$\le 2^{-pk^* - p/2}\varepsilon_+^*(G_n)$$

$$+ 2^{-p/2} \sum_{k=1}^{k^*} 2^{-p(k-1)} \sum_{Q_i^k \in \mathcal{Q}^k} |\varepsilon^*(Q_i^k)|.$$

Now, the proof is extremely similar to the previous one and by the same argument,

$$\mathbb{E} \sum_{Q \in \mathcal{Q}_k} |\varepsilon^*(Q)| \le 4^k \frac{2}{\sqrt{\pi}} 2^{\eta - k}\sigma. \qquad (81)$$

As in the previous proof,

$$\mathbb{E}\varepsilon_+^*(G_n) = \frac{n^2}{\sqrt{\pi}}\sigma. \qquad (82)$$

Altogether,

$$\mathbb{E}W_p^p(\varepsilon_+^*, \varepsilon_-^*) \qquad (83)$$

$$\le 2^{-pk^* - p/2} \frac{4^\eta}{\sqrt{\pi}}\sigma + 2^\eta 2^{p/2} \sum_{k=1}^{k^*} 2^{-(p-1)k} \frac{2}{\sqrt{\pi}}\sigma$$

$$\le 2^{-pk^* - p/2} \frac{4^\eta}{\sqrt{\pi}}\sigma + 2^\eta 2^{p/2} \frac{2}{\sqrt{\pi}}\sigma \left( \frac{1 - 2^{-(p-1)k^*}}{2^{p-1} - 1} \right).$$

We take $k^* = \eta$ again to get

$$\mathbb{E}W_p^p(\varepsilon_+^*, \varepsilon_-^*) \qquad (84)$$

$$\leq 2^{-(p-1)\eta - p/2} \frac{2^\eta}{\sqrt{\pi}} \sigma + \frac{2^\eta 2^{p/2+1}}{\sqrt{\pi}} \sigma \left( \frac{1}{2^{p-1}-1} \right)$$

$$\leq \frac{2^\eta \sigma}{\sqrt{\pi}} \left( 2^{-(p-1)\eta - p/2} + \frac{2^{(p+2)/2}}{2^{p-1}-1} \right).$$

Remark that $2^{-(p-1)\eta - p/2} \leq \sqrt{2}/2$ and that $\frac{2^{(p+2)/2}}{2^{p-1}-1}$ is decreasing with value 4 at 2. Thus the expression is bounded by $4 + \sqrt{2}/2 \leq 4\sqrt{2}$ and the claim follows. $\square$

**Lower bound for $W_p$ between noisy counterparts.** Let $\mu : G_n \to [0, 1]$ be a non-negative image on the $n \times n$ unit grid $G_n$, and let $\varepsilon_1, \varepsilon_2$ be independent random signed noise measures satisfying Assumption 3.2. For any integer $p \geq 1$, the expected signed $p$-Wasserstein discrepancy between the two noisy versions of the image is bounded from below by:

$$\mathbb{E}\left[ W_p^\pm(\mu + \varepsilon_1, \mu + \varepsilon_2) \right] \geq C_p \, n^{\frac{2}{p}-1} \sigma^{\frac{1}{p}}. \qquad (85)$$

Where $C_p = \frac{2^{\frac{1}{p}-1}}{\sqrt{\pi}} \Gamma\left( \frac{1}{2p} + \frac{1}{2} \right)$.

By definition, the signed Wasserstein distance is given by

$$W_p^\pm(\mu + \varepsilon_1, \mu + \varepsilon_2)$$
$$= W_p((\mu + \varepsilon_1)_+ + (\mu + \varepsilon_2)_-, (\mu + \varepsilon_2)_+ + (\mu + \varepsilon_1)_-).$$

Let $\varepsilon^* = \varepsilon_2 - \varepsilon_1$. At every pixel $x \in G_n$, the difference between the target and source mass is identically

$$(\mu + \varepsilon_2)_+(x) + (\mu + \varepsilon_1)_-(x)$$
$$\quad - ((\mu + \varepsilon_1)_+(x) + (\mu + \varepsilon_2)_-(x))$$
$$= \mu(x) + \varepsilon_2(x) - (\mu(x) + \varepsilon_1(x)) = \varepsilon^*(x)$$

Let $M$ be the total excess mass that must be moved. Because $\sum_x \varepsilon_1(x) = 0$ and $\sum_x \varepsilon_2(x) = 0$ (as established in Proposition 3.3), we have $\sum_x \varepsilon^*(x) = 0$, so the total excess equals the total deficit:

$$M = \sum_{x \in G_n} (\varepsilon^*(x))_+. \qquad (86)$$

While the overlap mass,

$$\min(((\mu+\varepsilon_2)_+ + (\mu+\varepsilon_1)_-)(x), ((\mu+\varepsilon_1)_+ + (\mu+\varepsilon_2)_-)(x))$$

can stay in place, any unit of the excess mass $M$ must be transported to a different pixel. Since the grid $G_n$ has a minimum inter-pixel distance of $1/n$, any mass that moves incurs a cost of at least $(1/n)^p$. Consequently, the total transport cost satisfies:

$$W_p^p((\mu+\varepsilon_1)_+ + (\mu+\varepsilon_2)_-, (\mu+\varepsilon_2)_+ + (\mu+\varepsilon_1)_-) \geq \left( \tfrac{1}{n} \right)^p M. \qquad (87)$$

Taking the $p$-th root yields $W_p^\pm(\mu + \varepsilon_1, \mu + \varepsilon_2) \geq M^{1/p}/n$. Because the function $t \mapsto t^{1/p}$ is concave for $p \geq 1$, we can apply the power mean inequality (or Jensen's inequality) to the sum of $m = n^2$ elements to yield:

$$M^{\frac{1}{p}} = \left( \sum_{i=1}^m (\varepsilon^*(x_i))_+ \right)^{\frac{1}{p}} \geq m^{\frac{1}{p}-1} \sum_{i=1}^m (\varepsilon^*(x_i))_+^{\frac{1}{p}}. \qquad (88)$$

Taking the expectation and recalling that $\varepsilon_1$ and $\varepsilon_2$ are independent with marginal distributions $\mathcal{N}(0, \sigma^2)$, the difference $\varepsilon^*$ has a marginal distribution of $\mathcal{N}(0, 2\sigma^2)$. Therefore,

$$\mathbb{E}\left[ M^{1/p} \right] \geq (n^2)^{\frac{1}{p}-1} \sum_{i=1}^{n^2} \mathbb{E}\left[ (\varepsilon^*(x_i))_+^{1/p} \right] \qquad (89)$$

$$= n^{\frac{2}{p}-2} \cdot n^2 \cdot \mathbb{E}\left[ (\varepsilon^*)_+^{1/p} \right] \qquad (90)$$

$$= n^{\frac{2}{p}} \mathbb{E}\left[ (\varepsilon^*)_+^{1/p} \right]. \qquad (91)$$

The expectation of the fractional power of the half-normal distribution is given by the standard absolute moments of the Gaussian distribution. Specifically, for $\varepsilon^* \sim \mathcal{N}(0, 2\sigma^2)$:

$$\mathbb{E}\left[ (\varepsilon^*)_+^{1/p} \right] = \frac{1}{2} \mathbb{E}\left[ |\varepsilon^*|^{1/p} \right] \qquad (92)$$

$$= (2\sigma^2)^{\frac{1}{2p}} \frac{2^{\frac{1}{2p}-1}}{\sqrt{\pi}} \Gamma\left( \frac{1}{2p} + \frac{1}{2} \right) = C_p \sigma^{1/p}.$$

Combining these results gives the final bound:

$$\mathbb{E}\left[ W_p^\pm(\mu + \varepsilon_1, \mu + \varepsilon_2) \right] \geq \frac{1}{n} \mathbb{E}\left[ M^{1/p} \right] \geq C_p n^{\frac{2}{p}-1} \sigma^{\frac{1}{p}}. \qquad (93)$$

**On the asymptotic gap and spatial geometry of noise.** While our lower bound captures the correct empirical $\sigma^{1/p}$ scaling, it leaves an asymptotic gap with respect to the grid resolution $n$. For $p = 2$, the lower bound scales as $\Omega(\sigma^{1/2})$, whereas Theorem 3.5 establishes an upper bound of $O(n^{1/2}\sigma^{1/2})$. This gap arises because our construction penalizes the must-move mass $M = \Theta(n^2 \sigma)$ by the minimum possible grid distance, $1/n$. For $p = 2$, the transport cost $(1/n)^2$ perfectly cancels the $n^2$ mass growth. In reality, noise often clusters spatially, forcing mass to travel distances strictly larger than $1/n$. Capturing this multiscale spatial transport from below, especially since $W_p^\pm$ lacks the triangle inequality for $p > 1$ remains an open challenge.

**Theorem 3.7** Let $\mu, \nu : G_n \to [0, 1]$ be two probability measures on the $n \times n$ unit grid $G_n$ with cyclic boundary conditions and let $\varepsilon_\mu, \varepsilon_\nu : G_n \to \mathbb{R}$ be signed noise measures that satisfy Assumption 3.2. For convenience we assume that $n = 2^\eta$, for $\eta \in \mathbb{N}$. Then

$$\mathbb{E}\left[ W_1^\pm(\mu + \varepsilon_\mu, \nu + \varepsilon_\nu) - W_1^\pm(\mu, \nu) \right] \qquad (94)$$

$$\leq \frac{4n \log_2 n + n}{\sqrt{\pi}} \sigma + \frac{\sqrt{2}}{n}. \qquad (95)$$

*Proof of Theorem 3.7.* Recall that $W_1^\pm$ satisfies the triangle inequality, so

$$W_1^\pm(\mu + \varepsilon_\mu, \nu + \varepsilon_\nu) \tag{96}$$
$$\leq W_1^\pm(\mu + \varepsilon_\mu, \mu) + W_1^\pm(\mu, \nu) + W_1^\pm(\nu, \nu + \varepsilon_\nu).$$

By symmetry

$$\mathbb{E}W_1^\pm(\mu + \varepsilon_\mu, \mu) = \mathbb{E}W_1^\pm(\nu, \nu + \varepsilon_\nu) \tag{97}$$

Therefore,

$$\mathbb{E}[W_1^\pm(\mu + \varepsilon_\mu, \nu + \varepsilon_\nu) - W_1^\pm(\mu, \nu)] \tag{98}$$
$$\leq 2\mathbb{E}W_1^\pm(\mu, \mu + \varepsilon_\mu).$$

We proceed to upper-bound the RHS. By the definition of the signed Wasserstein metric,

$$W_1^\pm(\mu, \mu + \varepsilon) = W_1(\mu_+ + (\mu + \varepsilon)_-, (\mu + \varepsilon)_+ + \mu_-)$$
$$= W_1(\mu + (\mu + \varepsilon)_-, (\mu + \varepsilon)_+). \tag{99}$$

Where the last equality follows from the fact that $\mu_+ = \mu$ and $\mu_- = 0$. We now use the dyadic upper bound in (12). The image is partitioned into 4 quadrants recursively, thus $\delta = 1/2$. Our domain has diameter $\sqrt{2}/2$ since it is the discrete $n \times n$ unit grid $G_n \subset [0,1] \times [0,1] \in \mathbb{R}^2$ with cyclic boundary conditions. The inequality only holds for probability measures, so we need to rescale.

$$W_1^\pm(\mu, \mu_\varepsilon) \tag{100}$$
$$= (\mu + \varepsilon)_+(G_n) \times W_1^\pm\left(\frac{\mu + (\mu + \varepsilon)_-}{(\mu + \varepsilon)_+(G_n)}, \frac{(\mu + \varepsilon)_+}{(\mu + \varepsilon)_+(G_n)}\right)$$
$$\leq \frac{\sqrt{2}}{2} \cdot 2^{-k^*}(\mu + \varepsilon)_+(G_n) + \frac{\sqrt{2}}{2}\sum_{k=1}^{k^*} 2^{-(k-1)}$$
$$\times \sum_{Q_i^k \in \mathcal{Q}^k} |(\mu + (\mu + \varepsilon)_-)(Q_i^k) - (\mu + \varepsilon)_+(Q_i^k)|.$$

By considering the two cases $(\mu + \varepsilon)(Q_i^k) \geq 0$ and $(\mu + \varepsilon)(Q_i^k) < 0$ it is easy to see that the term $(\mu + (\mu + \varepsilon)_-)(Q_i^k) - (\mu + \varepsilon)_+(Q_i^k)$ is equal to $-\varepsilon(Q_i^k)$, so the bound above simplifies to

$$W_1^\pm(\mu, \mu_\varepsilon) \leq 2^{-k^* - \frac{1}{2}}(\mu + \varepsilon)_+(G_n) \tag{101}$$
$$+ \frac{\sqrt{2}}{2}\sum_{k=1}^{k^*} 2^{-(k-1)}\sum_{Q_i^k \in \mathcal{Q}^k} |\varepsilon(Q_i^k)|. \tag{102}$$

Recall that $Q_i^k$ is a square region of size $2^{\eta-k} \times 2^{\eta-k}$. Then, $\varepsilon(Q_i^k)$ is a sum of negatively correlated random variables so that

$$\text{Var}\,\varepsilon(Q_i^k) \leq |Q_i^k|\sigma^2.$$

Recall that $\mathbb{E}|X| = \sigma\sqrt{2/\pi}$ when $X \sim \mathcal{N}(0, \sigma^2)$. Thus,

$$\mathbb{E}|\varepsilon(Q_i^k)| \leq \sqrt{\frac{2}{\pi}}\sigma 2^{\eta-k}. \tag{103}$$

Summing over the $4^k$ cells at level $k$,

$$\mathbb{E}\sum_{Q \in \mathcal{Q}_k} |\varepsilon(Q)| \leq 4^k\sqrt{\frac{2}{\pi}}\sigma 2^{\eta-k}. \tag{104}$$

Plugging this back into the RHS of (101) and recalling that $2^\eta = n$ gives

$$\mathbb{E}\left[\frac{\sqrt{2}}{2}\sum_{k=1}^{k^*} 2^{-(k-1)}\sum_{Q_i^k \in \mathcal{Q}^k} |\varepsilon(Q_i^k)|\right] \tag{105}$$
$$\leq \frac{\sqrt{2}}{2}\sum_{k=1}^{k^*} 2^{-(k-1)}4^k\sqrt{\frac{2}{\pi}}\sigma 2^{\eta-k}$$
$$= \frac{2^{\eta+1}\sigma}{\sqrt{\pi}}k^*.$$

We take $k^* = \eta = \log_2 n$ to obtain the bound

$$\mathbb{E}W_1^\pm(\mu, \mu_\varepsilon) \leq \frac{1}{\sqrt{2n}}\mathbb{E}[(\mu + \varepsilon)_+(G_n)] + \frac{2n\log_2 n}{\sqrt{\pi}}\sigma. \tag{106}$$

We now bound the first term in the RHS.

$$\mathbb{E}[(\mu + \varepsilon)_+(G_n)] \leq \mathbb{E}[\mu_+(G_n)] + \mathbb{E}[\varepsilon_+(G_n)] \tag{107}$$
$$= 1 + \mathbb{E}[\varepsilon_+(G_n)]$$

where the last equality follows from the fact that $\mu$ is a (non-negative) probability measure. By a symmetry argument

$$\mathbb{E}\varepsilon_+(G_n) = \tfrac{1}{2}\mathbb{E}|\varepsilon|(G_n). \tag{108}$$

Set $\mathfrak{s} = \frac{1}{2}\sum_x |\varepsilon(x)|$ and recall that $\varepsilon(x) \sim \mathcal{N}(0, \sigma^2)$ to establish,

$$\mathbb{E}\mathfrak{s} = \frac{n^2}{2}\sqrt{\frac{2}{\pi}}\sigma. \tag{109}$$

Thus,

$$\frac{1}{\sqrt{2n}}\mathbb{E}[(\mu + \varepsilon)_+(G_n)] \leq \frac{1}{\sqrt{2n}} + \frac{\sigma}{2\sqrt{\pi}}n. \tag{110}$$

Plugging this back into (106) gives

$$\mathbb{E}W_1^\pm(\mu, \mu_\varepsilon) \leq \frac{2n\log_2 n + n/2}{\sqrt{\pi}}\sigma + \frac{1}{\sqrt{2n}}. \tag{111}$$

Note that the same bound applies to $\mathbb{E}W_1^\pm(\nu, \nu + \varepsilon_\nu)$. By subtracting $W_1^\pm(\mu, \nu)$ from both sides of (96) and taking expectations, we have

$$\mathbb{E}\left[W_1^\pm(\mu_\varepsilon, \nu_\varepsilon) - W_1^\pm(\mu, \nu)\right] \tag{112}$$
$$\leq \mathbb{E}W_1^\pm(\mu_\varepsilon, \mu) + \mathbb{E}W_1^\pm(\nu, \nu_\varepsilon)$$
$$\leq \frac{4n\log_2 n + n}{\sqrt{\pi}}\sigma + \frac{\sqrt{2}}{n}. \qquad \square$$

**Theorem 3.8** Let $\mu, \nu : G_n \to [0, 1]$ be two probability measures on the $n \times n$ unit grid $G_n$ with cyclic boundary conditions and let $\varepsilon_\mu, \varepsilon_\nu : G_n \to \mathbb{R}$ be signed noise measures that satisfy Assumption 3.2. For convenience we assume that $n = 2^\eta$, for $\eta \in \mathbb{N}$. Then

$$\mathbb{E}\big[W_p^\pm(\mu + \varepsilon_\mu, \nu + \varepsilon_\nu)\big] \tag{113}$$

$$\leq \left(\tfrac{\sqrt{2}}{2}\right)^{1-\frac{1}{p}} W_1(\mu, \nu)^{\frac{1}{p}} + \tfrac{\sqrt{2}}{2}\left(\tfrac{4}{\sqrt{\pi}} n \log_2 n + \tfrac{1}{\sqrt{\pi}} n\right)^{\frac{1}{p}} \sigma^{\frac{1}{p}}.$$

*Proof of Theorem 3.8.* Using Proposition A.4

$$\mathbb{E}\big[W_p^\pm(\mu + \varepsilon_\mu, \nu + \varepsilon_\nu)\big] \tag{114}$$
$$\leq \mathbb{E}\big[D^{1-\frac{1}{p}}\big(W_1(\mu, \nu) + W_1(\varepsilon_+^*, \varepsilon_-^*)\big)^{\frac{1}{p}}\big]$$

The function $t \mapsto t^{1/p}$ is concave on $[0, \infty)$, thus

$$\mathbb{E}\big[D^{1-\frac{1}{p}}\big(W_1(\mu, \nu) + W_1(\varepsilon_+^*, \varepsilon_-^*)\big)^{\frac{1}{p}}\big] \tag{115}$$
$$\leq D^{1-\frac{1}{p}}\big(\mathbb{E}\big[W_1(\mu, \nu) + W_1(\varepsilon_+^*, \varepsilon_-^*)\big]\big)^{\frac{1}{p}}.$$

By the linearity of expectation,

$$D^{1-\frac{1}{p}}\big(\mathbb{E}\big[W_1(\mu, \nu) + W_1(\varepsilon_+^*, \varepsilon_-^*)\big]\big)^{\frac{1}{p}} \tag{116}$$
$$= D^{1-\frac{1}{p}}\big(W_1(\mu, \nu) + \mathbb{E}\big[W_1(\varepsilon_+^*, \varepsilon_-^*)\big]\big)^{\frac{1}{p}}$$

Finally, using Theorem 3.4 we get that

$$D^{1-\frac{1}{p}}\big(W_1(\mu, \nu) + \mathbb{E}\big[W_1(\varepsilon_+^*, \varepsilon_-^*)\big]\big)^{\frac{1}{p}} \tag{117}$$
$$\leq D^{1-\frac{1}{p}}\left(W_1(\mu, \nu) + \tfrac{2\sqrt{2}}{\sqrt{\pi}}\sigma n \log_2 n + \tfrac{1}{\sqrt{2\pi}}\sigma n\right)^{\frac{1}{p}}.$$

Using Jensen,

$$\mathbb{E}\big[W_p^\pm(\mu + \varepsilon_\mu, \nu + \varepsilon_\nu)\big] \leq (\tfrac{\sqrt{2}}{2})^{1-\frac{1}{p}} W_1(\mu, \nu)^{\frac{1}{p}}$$
$$+ \tfrac{\sqrt{2}}{2}\left(\tfrac{4}{\sqrt{\pi}} n \log_2 n + \tfrac{1}{\sqrt{\pi}} n\right)^{\frac{1}{p}} \sigma^{\frac{1}{p}}. \tag{118}$$

$\square$

# B. Unbalanced optimal transport

Various approaches have been proposed to generalize the idea of optimal transport to the case of two measures whose total mass is not equal. See Caffarelli & McCann (2010); Liero et al. (2018); Figalli (2010), for instance. Among these proposals, one is particularly amenable to the analysis we carried out. Given $\mu, \nu \in \mathcal{M}_+(X)$ two positive measures on a set $X$ that do not necessarily have the same mass, the set of *subcouplings* of $\mu$ and $\nu$ is defined as

$$\Gamma_\leq(\mu, \nu) := \{\pi \in \mathcal{M}_+(X \times X) : \pi(A \times X) \tag{119}$$
$$\leq \mu(A), \pi(X \times B)$$
$$\leq \nu(B), \text{ for all } A, B \in \mathcal{B}(X)\},$$

where $\mathcal{B}(X)$ is the set of Borel measures on $X$. For simplicity, set $m_\mu := \mu(X)$, $m_\nu := \nu(X)$ and $m_\pi := \pi(X \times X)$. Then, the $(p, C)$ unbalanced Kantorovich–Rubinstein distance is defined by

$$\mathrm{KR}_{p,C}(\mu, \nu) := \left(\inf_{\pi \in \Gamma_\leq(\mu, \nu)}\left[\int_{X \times X} d^p(x, y)\, \mathrm{d}\pi(x, y)\right.\right.$$
$$\left.\left. + C^p\left(\tfrac{m_\mu + m_\nu}{2} - m_\pi\right)\right]\right)^{\frac{1}{p}} \tag{120}$$

The parameter $C$ determines the range of admissible transport. Indeed, any subcoupling transferring mass between points that are further apart than $C$ cannot be optimal, as destroying the mass would lead to a smaller objective function.

**Proposition B.1.** *Consider a square $2^\eta \times 2^\eta$ grid, where $\eta \geq 0$ is integer, and a dyadic partition scheme. Let $\mu, \nu$ be two measures on the grid, not necessarily with equal masses. It holds that,*

$$\mathrm{KR}_{p,C}(\mu, \nu) \leq \frac{C^p}{2}|m_\mu - m_\nu| \tag{121}$$
$$+ \mathrm{diam}(S)^p 2^{3p-1} \sum_{k=\ell^*}^{\eta} 2^{-pk} \sum_{Q \in \mathcal{D}^k} |\mu(Q) - \nu(Q)|.$$

*where $\ell^* = 1 + \min\big(L, \lfloor \max\big(0, \log_2\big(2\,\mathrm{diam}(S)/C\big)\big)\rfloor\big)$.*

*Proof.* Looking at the objective in (120), a strategy to construct a good subcoupling is to match as much mass below scale $C$ as possible and then just pay $C^p$ for the mass that hasn't been coupled. Because of the coarse-to-fine dyadic decomposition, each pixel is a final leaf of the decomposition tree.

One can then apply Lemma 3.15 in Struleva et al. (2026) giving bounds on the distance on trees. $\square$

Similarly to the above, we can define

$$\mathrm{KR}_{p,C}^\pm(\mu, \nu) := \mathrm{KR}_{p,C}(\mu_+ + \nu_-, \nu_+ + \mu_-). \tag{122}$$

**Theorem B.2.** *Let $\mu : G_n \to [0, 1]$ be a probability measure on the $n \times n$ unit grid $G_n$ with cyclic boundary conditions, and let $\varepsilon$ satisfy Assumption 3.2. Further assume that $n = 2^\eta$, for $\eta \in \mathbb{N}$. Then, for $C > 0$,*

$$\mathbb{E}\,\mathrm{KR}_{p,C}^\pm(\mu + \varepsilon, \mu) \tag{123}$$
$$\leq \begin{cases} \mathrm{diam}(S)^p 2^{3p-1} \tfrac{\sqrt{2}\sigma}{\sqrt{\pi}} n(\eta - \ell^* + 1) & \text{if } p = 1, \\ \mathrm{diam}(S)^p 2^{3p-1} \tfrac{\sigma}{\sqrt{2\pi}} n(2^{2-\ell^*} - 2^{1-\eta}) & \text{if } p = 2. \end{cases}$$

*Proof of Theorem B.2.* We apply Proposition B.1 to the probability measures $\mu_- + (\mu + \varepsilon)_+$ as well as $(\mu + \varepsilon)_- + \mu_+$.

First, note that

$$\mathbb{E}|m_{\mu_-+(\mu+\varepsilon)_+} - m_{\mu_++(\mu+\varepsilon)_-}| \tag{124}$$

$$= \mathbb{E}\left|\sum_x (\mu_- - \mu_+)(x) + (\mu+\varepsilon)_+(x) - (\mu+\varepsilon)_-(x)\right| = 0.$$

Taking expectations and using independence and zero mean of the noise,

$$\sum_{k=\ell^*}^{\eta} 2^{-pk} \sum_{Q\in\mathcal{D}^k} |\mu_-(Q) + (\mu+\varepsilon)_+(Q) \tag{125}$$

$$- (\mu+\varepsilon)_-(Q) - \mu_+(Q)| = \sum_{k=\ell^*}^{\eta} 2^{-pk} \sum_{Q\in\mathcal{D}^k} |\varepsilon(Q)|.$$

Since each $\varepsilon(x)$ is Gaussian with variance $\sigma^2$, one has $\mathbb{E}|\sum_{x\in Q} \varepsilon(x)| \le \sigma\sqrt{|Q|}\sqrt{2/\pi}$. Furthermore, the dyadic family $\mathcal{D}_k$ has $|\mathcal{D}_k| = 2^{2k}$ cubes of cardinality $|Q| = n^2/2^{2k}$. Therefore

$$\sum_{Q\in\mathcal{D}_k} \mathbb{E}\left|\sum_{x\in Q} \varepsilon(x)\right| \le \sigma\sqrt{\tfrac{2}{\pi}} \sum_{Q\in\mathcal{D}_k} \sqrt{|Q|} \tag{126}$$

$$= \sigma\sqrt{\tfrac{2}{\pi}} \cdot 2^{2k} \cdot \frac{n}{2^k} = \sigma\sqrt{\tfrac{2}{\pi}}\, n\, 2^k.$$

Plugging this into the multiscale sum yields

$$\sum_{k=\ell^*}^{\eta} 2^{-pk} \sum_{Q\in\mathcal{D}_k} \mathbb{E}\left|\sum_{x\in Q} \varepsilon(x)\right| \le \sigma\sqrt{\tfrac{2}{\pi}}\, n \sum_{k=\ell^*}^{\eta} 2^{(1-p)k}. $$
$$\tag{127}$$
$$\square$$

