# OpenReview forum: "Quantifying the noise sensitivity of the Wasserstein metric for images"
_ICML.cc/2026/Conference — ICML 2026 regular_

### Official Review · Reviewer_J1kj · 2026-02-16

**Soundness:** 3
**Presentation:** 4
**Significance:** 2
**Originality:** 3
**Overall Recommendation:** 4
**Confidence:** 3

**Summary:**

The paper provides theoretical bounds for the sensitivity of the Wasserstein metric of different orders to gaussian noise on images. The paper shows that Wasserstein is more robust than L2 Euclidean distance, and that W2 is more robust than W1. This is shown both theoretically and empirically on Cryo-EM images.

**Compliance With Llm Reviewing Policy:**

Affirmed.

**Final Justification:**

I would provide a weak-accept rating here.

My main concern previously was the significance of this work. Namely: How accurate is the Gaussian noise model, How wide are the bounds, and if W2 is indeed a preferred metric over W1 in real use cases.

The authors have provided enough evidence from the literature that the Gaussian noise model is accurate. This corresponds with the other literature I have read.

The final provided figures also provide some additional evidence that W2 is a more useful metric than W1, which makes the significance of this paper greater.

The bounds provided are still quite wide though. Even if W1 is more useful than L2 distance, the issue is that the derived bounds do not show this. I think this is still a flaw of the overall paper. However, because the paper is overall well-written, seems technically sound, and still makes progress on the problem of understanding the behavior of popular metrics, I would lean towards acceptance.

**Key Questions For Authors:**

1) How restrictive/realistic is Assumption 3.2, specifically the distribution of the noise?

2) Would a lower bound on W2 be possible? Referring to Theorem 3.5 only providing an upper bound for Wp.

3) As mentioned in the paper's conclusion: Knowing that W2 is more robust to noise can be helpful, but this also assumes that W2 measures all features of importance for a given use-case over W1. Can you demonstrate further perhaps that W2 provides a useful measure of image similarity for use-cases like CryoEM? The fact that W2 does not have the same metric properties as W1 does give me some pause about its interpretability for experimentalists.

4) Can you provide additional explanation for Figure 8? Perhaps axis labels and a color bar would help make clear what the figure is showing?

**Limitations:**

missing impact statement

**Strengths And Weaknesses:**

I felt that this paper was well-written, and it was easy to follow the overall story.

I appreciated how the paper provided significant relevant background, and felt that the paper did a great job of experimentally validating all theory presented. It was very easy to understand the significance of each experiment, and which theorem it was supporting.

My main concerns about this paper are in the significance of the results, and the restrictive nature of the assumptions.

- The paper assumes zero-mean Gaussian noise only. This is a quite limited use-case for most imaging applications, where noise distributions are far more complex (e.g. Poisson) or pixel values are clipped. Additional discussion about how realistic this assumption is would be important for understanding the practical use of these bounds.

- Theorem 3.5 does not provide a lower bound for $W_p$.

- As the paper mentions, the observed bounds are quite wide. For example, in Figure 3, it seems as if the sensitivity of the L2 metric would fit within the bounds provided for W1.

In general, after reviewing the paper, I believe the paper is technically sound, although its significance is limited.

One small note:

Figure 3 is referenced before Figure 2 in the main text.

---

> ### Author Rebuttal · Authors · 2026-03-31
>
> Thank you for the constructive feedback. A point-by-point response follows:
>
> **Question 1:**
> How restrictive/realistic is Assumption 3.2, specifically the distribution of the noise?
>
> **Response:**
> We believe that it is a very good model for the application domains we consider. In the Cryo-EM modality, Poisson shot noise is typically considered to be the most physically-faithful model and is well-approximated by a Gaussian distribution. Likewise, in medical imaging Gaussian noise modeling is standard. Originally we wanted to analyze the classic i.i.d. Gaussian model. However, this would have resulted in a very slight mass-imbalance even when the original (noiseless) images have equal masses. There are two common ways to resolve mass imbalance in optimal transport:
> 1. Use a method for unbalanced OT which includes a penalty term for mass imbalance.
> 1. Normalize the noisy images so that they will have equal masses.
>
> The second choice, of normalizing by mean-subtraction, results in the noise model we have chosen to present in the main paper. We were happy to see that this choice resulted in short and elegant proofs. The analysis of unbalanced transport appears in Appendix B. We can place it in the main paper, place permitting. However, we think that the added complexity of this might be off-putting to some practitioners.
>
> It is certainly possible to extend the analysis to more general noise models. Please see our response to Reviewer EzAq.
>
> **Question 2:** Would a lower bound on $W_2$ be possible? Referring to Theorem 3.5 only providing an upper bound for $W_p$.
>
> **Response:**
> It is possible to derive lower bounds, which are unfortunately not sharp. The main difficulty comes from the intricate nature of c-transforms in optimal transport. If one attempts strategies based on metric comparison, our computations suggest that sliced OT loses in accuracy, while dual Sobolev norms require strong assumptions on the signal.
> This being said,  we have achieved a non-sharp lower bound for $W_p$ and added it into the existing manuscript. The lower bounds scale similarly with the power of $\sigma$ (p-th root of the noise). We will include both the formal proof and the updated simulation results in the camera-ready version.
>
> **Question 3:** As mentioned in the paper's conclusion: Knowing that $W_2$ is more robust to noise can be helpful, but this also assumes that $W_2$ measures all features of importance for a given use-case over $W_1$. Can you demonstrate further perhaps that $W_2$ provides a useful measure of image similarity for use-cases like CryoEM? The fact that $W_2$ does not have the same metric properties as $W_1$ does give me some pause about its interpretability for experimentalists.
> ** Response:**
> See answer to next question.
>
> **Question 4:** Can you provide additional explanation for Figure 8? Perhaps axis labels and a color bar would help make clear what the figure is showing?
>
> **Response:**
> We acknowledge that this figure wasn't clear enough. We have improved that section, observing the following. Under purely additive noise, $W_1$ and $W_2$ yield highly correlated relative distance rankings, which creates nearly identical visual gradients and masks $W_2$’s theoretically favorable scaling. To show $W_2$'s practical superiority, we must exploit its quadratic cost, which heavily penalizes actual spatial shifts while forgiving local noise fluctuations. To this end we ran a new simulation and generated new plots. For more details, please see our response to Question 1 of Reviewer EzAq.
>
> **Missing impact statement:** The missing impact statement will be included in the camera-ready version.
>
> ===
>
> We kindly ask the reviewer to reconsider their overall recommendation in light of our responses and also because the recommendation to reject seems incongruent with the rest of the review which gave high scores for soundness (excellent) and presentation (excellent).

---

> > ### Author Rebuttal · Reviewer_J1kj · 2026-04-01
> >
> > Thank you for your response.
> >
> > I have adjusted my rating based on the response to my review and other reviews. I am willing to adjust my rating further with some additional clarifications.
> >
> > Q1: Can you provide citations for your argument that the Gaussian noise model is accurate? I appreciate the clarification that you can also handle others, but I would like to verify that your claim is accurate.
> >
> > In response to Q2, Q3, and Q4, you mention the presence of new simulation plots. The results, as described, would help me understand better the significance of this work. Can you provide an anonymous link to these figures? This is allowed under ICML guidelines and would help verify the claims you have provided in your response.
> >
> > For the plots around bounds for $W_p$, I am particularly interested in understanding how wide the bounds are. For example in Figure 3, the bounds for $W_1$ seem to contain the L2 distance already.
> >
> > Thank you again for your response.

---

> > > ### Author Response · Authors · 2026-04-07
> > >
> > > We thank the reviewer engaging with us and adjusting their score based on the discussion. Here are our point-by-point answers:
> > >
> > > **Question 1**: Can you provide citations for your argument that the Gaussian noise model is accurate? I appreciate the clarification that you can also handle others, but I would like to verify that your claim is accurate.
> > >
> > > **Answer 1**: In the context of cryo-EM, here is what the review article [1] that we cite in our manuscript says:
> > > “The chief noise source in cryo-EM at the frame level (before motion correction) is shot noise, which follows a Poisson distribution. After movie frames are averaged to produce micrographs, it is customary to assume that the noise is characterized by a Gaussian distribution. Indeed, all current algorithms build on—implicitly or explicitly—the speculated Gaussianity of the noise.”
> > >
> > > We note briefly that SOTA approaches to reconstruction in cryo-EM assume a non-white noise model (where different frequency bands have different noise levels) which is estimated and accounted for in downstream processing, for example by whitening the signal to produce Gaussian i.i.d. pixel noise.
> > >
> > > [1] T. Bendory, A. Bartesaghi and A. Singer, "Single-Particle Cryo-Electron Microscopy: Mathematical Theory, Computational Challenges, and Opportunities," in IEEE Signal Processing Magazine, vol. 37, no. 2, pp. 58-76, March 2020, doi: 10.1109/MSP.2019.2957822.
> > >
> > > **Question 2**: In response to Q2, Q3, and Q4, you mention the presence of new simulation plots. The results, as described, would help me understand better the significance of this work. Can you provide an anonymous link to these figures? This is allowed under ICML guidelines and would help verify the claims you have provided in your response.
> > >
> > > **Answer 2**:Please find the plot discussed here: https://imgur.com/a/mm7lU2i
> > > The plot aims at showing the robustness of $W_2$ to noise compared to the other metrics. In these plots, the intersection between the dotted line and the full line of the same color is where the metric fails to distinguish between the shift and the noise.
> > > As we discussed in the previous comment, this shows that $W_2$ tolerates ~3.5x more noise that $W_1$ before it starts confusing additive noise with a genuine spatial shift.
> > >
> > > **Question 3**: For the plots around bounds for , I am particularly interested in understanding how wide the bounds are. For example in Figure 3, the bounds for  seem to contain the L2 distance already.
> > >
> > > **Answer 3**: Please find the plot discussed here: https://imgur.com/a/rz0vC8C
> > > The updated figure 3 will include the 95% CI and the lower bounds, as we have mentioned in the response to reviewer jTPm. You are correct that the expected value of the $L_2$ distance numerically falls within the bounds for $W_1$. However, this numerical overlap does not imply that the metrics capture the same information or behave similarly in practice. As demonstrated in fig 8 of the paper, both Wasserstein metrics are superior to $L_2$ in the given context.

---

### Official Review · Reviewer_jTPm · 2026-03-07

**Soundness:** 3
**Presentation:** 3
**Significance:** 3
**Originality:** 3
**Overall Recommendation:** 4
**Confidence:** 4

**Summary:**

This paper proposes to study how the Wasserstein metric can be used to quantify the (additive Gaussian) noise in a image, when a grayscale image is modeled as a signed measure supported on a grid, and the intensity of a pixel represents its mass (which can be negative, as in cryo-EM). In particular, it provides insight into why the Wasserstein distance can be advantageous in high noise setting, as it is more robust to noise than the Euclidean norm.

**Compliance With Llm Reviewing Policy:**

Affirmed.

**Final Justification:**

This paper presents an original approach to quantifying the noise in images using signed Wasserstein divergences $W_p^\pm$, divergences which remain understudied in the literature, especially within the field of image processing. While the experimental results are promising, a deeper understanding on the robustness of the method as $p$ increases would have been appreciated. On another hand, I appreciate the connection the authors made with reference [R1] in the discussion period. Emphasizing the unbalanced case would greatly strengthen the paper, in particular in terms of experiments, even if I recognize that this addition may be limited by page constraints. For these reason, I maintain my score as a "Weak accept".

[R1] B. Piccoli, F. Rossi and M. Tournus. A Wasserstein norm for signed measures, with application to nonlocal transport equation with source term. Communications in Mathematical Sciences, pp.1279-1301, vol.21, n.5 (2023).

**Key Questions For Authors:**

1. $W_1$ has not been defined between measures of different masses, still it is used in Proposition A.3. Have I missed something?

2. Is the variance across experiments high (e.g. across the 100 experiments of Figure 1)?

3. With regard to Figure 7, can we conjecture about the types of input images/noise for which the quantification of noise using $W_2$ would behave monotonically?

4. This final remark is intended more as a comment motivated by curiosity. The case where the signed measures do not share the same total mass is addressed in [R1], where they propose a Generalized Wasserstein distance extended to signed measures. It would be interesting to know whether your results might extend naturally to this framework and if the Generalized Wasserstein distance is consistent with Theorem 3.1.

[R1] B. Piccoli, F. Rossi and M. Tournus. A Wasserstein norm for signed measures, with application to nonlocal transport equation with source term. Communications in Mathematical Sciences, pp.1279-1301, vol.21, n.5 (2023).

**Limitations:**

Yes

**Strengths And Weaknesses:**

**Soundness :** This paper is globally sound both on the theoretical analysis and the experimental results. Still, in my opinion, because of the non-monotonicity of the $W_p, p\geq2$ with respect to the noise, care must be taken when stating that the theory suggests the Wasserstein distance is more robust as $p$ increases (see Conclusion). I also list below some (unsorted) minor comments.

1. There is a typo line 80, left column, : it should be $\phi^d$.

2. I might be wrong but line 108, right column, isn't the "expected squared $L^2$ distance between a signal and its noisy version" $n^2\sigma^2$?

3. In equation (13), it should be $W_1$, and not $W_1^\pm$.

4. In equation (60), it should be $\mathcal{N}(0,2\sigma^2 N)$ and not $\mathcal{N}(0,2\sigma^2 N^2)$. The confusion between $N$ and $n$ follows until the end of the proof.

5. In Proposition 3.3, I would clarify that the result 2 is true almost surely.

6. There is a squared root missing on the 2 in equation (77).

7. Some elements of the proofs are clarified later in the text, which makes the reading less fluid. For example, the expected absolute value of a Gaussian random variable is used in the proof of Theorem 3.4, line 725, left column, but this result is given later in the same proof, line 728, right column. This result is also clearly stated later on in the text, in the proof of Theorem 3.7. Another example is the value chosen for the parameter $\delta$ for the dyadic partition and the value of the diameter of the grid, which are given in the proof of Theorem 3.7, but also used in the proof of Theorem 3.4.

8. The sentence line 340, right column, is not finished.

**Presentation :** Overall, the paper is very well written and its goal and contributions are clearly stated. However, in my opinion, the following two points could be addressed :

1. The first paragraph of "Related work" appears somewhat disconnected from the core results of the paper. Conversely, relevant literature on optimal transport for image processing seems to be missing.

2. I am not sure whether the paragraph "Contrast with perturbed point cloud model" is entirely pertinent as these results appear quite different from those considered in the paper. In particular, since the focus is on images, it may not be necessary to go into so much detail in this part.

**Significance :** This work quantifies the additive noise in a image in terms of Wasserstein distance for signed measures, which is of particular interest in the image processing literature. Additionally, it discusses the non-monotonicity of the $W_2$ metric with respect to the noise (see e.g. Figure 5), an aspect that is relevant for practitioners who aim to use the Wasserstein distance in such settings.

**Originality :** The originality of this paper stems from (i) its focus on the signed Wasserstein distance, a divergence which remains largely unexplored in the existing literature, especially in image processing, and (ii) the quantification of noise in a image with respect to Wasserstein metrics.

---

> ### Author Rebuttal · Authors · 2026-03-31
>
> Thank you for the thorough review and valuable feedback. We have made several improvements to the paper in light of your comments and address your specific questions below.
>
> **minor comment 2:** I might be wrong but line 108, right column, isn't the "expected squared  $L^2$ distance between a signal and its noisy version"  $n^2 \sigma^2$ ?
>
> **Response:** Indeed, it is the classical distance that scales as we announced, not its squared version. This, along with all the other minor comments, will be fixed in the camera-ready version.
>
> **Presentation comment 2:** I am not sure whether the paragraph "Contrast with perturbed point cloud model" is entirely pertinent…
>
> **Response:** We agree with you. This paragraph however arose from a previous review round in which the reviewer apparently had difficulties understanding the difference between the two models.
>
> **Question 1:** $W_1$ has not been defined between measures of different masses, still it is used in Proposition A.3. Have I missed something?
>
> **Response:** The objects in A.3 do indeed have the same mass according to the definition of the noise model and the fact that the original image masses are equal. This can be understood by seeing that both objects are the positive and the negative parts of a measure summing up to 0.
>
> **Question 2:** Is the variance across experiments high (e.g. across the 100 experiments of Figure 1)?
>
> **Response:** We have changed the plot to include 95% simulation intervals. We hope that this answers the concern.
>
> **Question 3:** With regard to Figure 7, can we conjecture about the types of input images/noise for which the quantification of noise using $W_2$ would behave monotonically?
>
> **Response:** We conjecture that the effect is more pronounced for images where mass has to be transported far away, across regions with small intensity.
> In those cases, the additional noise in the otherwise empty regions can be used to reduce the matching cost.
>
> **Question 4:** This final remark is intended more as a comment motivated by curiosity. The case where the signed measures do not share the same total mass is addressed in [R1], where they propose a Generalized Wasserstein distance extended to signed measures. It would be interesting to know whether your results might extend naturally to this framework and if the Generalized Wasserstein distance is consistent with Theorem 3.1.
>
> **Response:** Thank you very much for pointing this reference to us. Being (apparently) from slightly different communities, we didn’t know about that work; it shall be added to the references. Remark that our unbalanced approach in Appendix B requires constructing a subcoupling between the recombined positives and negative parts of the original measures. In the paper you brought to our attention, Lemma~12 highlights the existence of subcouplings that are minimizers of the Generalized Wasserstein distance. Up to constants, it seems that we already covered the concept you suggested without knowing it.

---

> > ### Author Rebuttal · Reviewer_jTPm · 2026-04-02
> >
> > I thank the authors for their clear responses, and for the nicely connection they made with the reference [R1]. However, I will maintain my original score, as my main concern remains: the claim that $W_p,\  p\geq 2$ becomes more robust to the noise as $p$ increases is undermined by the non-monotonicity of the function.

---

### Official Review · Reviewer_8uCT · 2026-03-10

**Soundness:** 3
**Presentation:** 3
**Significance:** 3
**Originality:** 3
**Overall Recommendation:** 5
**Confidence:** 4

**Summary:**

Overall, the article's critical contribution pertains to the theoretical analysis of noise sensitivity of Wasserstein metrics for images treated as discrete measures on pixel grids. The manuscript strives to address an important area in optimal transport: understanding how Wasserstein distances behave under pixel-wise additive noise. The main theoretical results are finite-sample expectation bounds for Gaussian noise models, proving that the signed 2-Wasserstein distance scales with the square root of noise standard deviation (favorable compared to Euclidean metric's linear scaling). The paper also identifies a peculiar phenomenon where increasing noise can decrease the Wasserstein distance, explained by noise "bridging" transport between structures. A cryo-EM case study demonstrates practical relevance in high-noise settings.

**Compliance With Llm Reviewing Policy:**

Affirmed.

**Key Questions For Authors:**

1. How do the bounds scale for larger images (e.g., 256x256 or 512x512)?
2. Have you compared with entropically regularized Wasserstein distances, which are more commonly used in practice?
3. Can you validate the findings on real cryo-EM data?
4. Is the code available for reproducing the experiments?

**Limitations:**

yes

**Strengths And Weaknesses:**

1. Soundness

Strengths:
- The theoretical analysis is rigorous and well-presented
- The multiscale approach (dyadic partition) is appropriate for obtaining sharp bounds
- The noise model (Assumption 3.2) with zero-sum constraint is clever and simplifies analysis
- The distinction from perturbed point cloud models (Section 2) is clearly articulated
- Quantitative validation of noise scaling (Figure 3) confirms theoretical predictions
- Visual comparison of metric robustness (Figure 1) clearly illustrates the advantage of W2
- Cryo-EM case study (Section 4.3) demonstrates practical relevance in a high-noise application
- Analysis of different image classes (Figure 6) shows the generality of findings

Weaknesses:
- The bounds in Theorem 3.8 appear loose (as acknowledged by the authors)
- The assumption of cyclic boundary conditions simplifies analysis but may not reflect real-world image boundaries
- The signed Wasserstein for p > 1 is not a metric (no triangle inequality), which limits its practical utility
- Limited to 32x32 images from DOTMark dataset—how do results scale to larger images?
- The cryo-EM experiment uses simulated data; validation on real cryo-EM images would strengthen claims
- No comparison with other robust metrics (e.g., Sinkhorn distance, sliced Wasserstein)

2. Presentation

Strengths:
- The paper is exceptionally well-written and clear
- Mathematical notation is precise and consistent
- The distinction between different problem settings (signed measures vs. point clouds) is clearly explained
- Figures effectively illustrate key concepts and results

Weaknesses:
- Some proofs are deferred to appendix without sketch in main text
- The "decreasing distance phenomenon" could be highlighted more prominently as a key contribution

3. Originality and Significance

Strengths:
- First systematic study of noise robustness of Wasserstein metrics for measures on fixed grids
- Novel theoretical bounds (Theorems 3.4-3.8) with explicit scaling laws
- The "decreasing distance phenomenon" (Section 4.4) is an interesting and counter-intuitive finding with a clear explanation
- The zero-sum noise model (Assumption 3.2) is well-motivated and avoids rescaling complications

Weaknesses:
- The dyadic bound (Proposition 2.1) is adapted from Weed & Bach (2019), not novel
- The signed Wasserstein construction follows Mainini (2012)

---

> ### Author Rebuttal · Authors · 2026-03-31
>
> We are grateful for the reviewer's detailed feedback and constructive suggestions. We have incorporated several revisions to strengthen the manuscript, which we address below.
>
> **Weakness:** Some proofs are deferred to appendix without sketch in main text
>
> **Response:** We will add proof sketches to the main text if there is enough room in the main text.
>
> **Weakness:**
> The assumption of cyclic boundary conditions simplifies analysis but may not reflect real-world image boundaries.
>
> **Response:**
> Our results can be easily adapted to other boundary conditions with a simple change in the resulting constants since the dyadic partitioning scheme still works.
>
> **Question 1:** How do the bounds scale for larger images (e.g., 256x256 or 512x512)?
>
> **Response:**
> We understand the bounds to mean the theoretical bounds, which provide the scaling as a function of the grid size. Therefore, we think that the reviewer meant empirical sample complexity. We could add a plot of the distance as a function of the image size. Is this what you meant in the question?
>
> **Question 2:** Have you compared with entropically regularized Wasserstein distances, which are more commonly used in practice?
>
> **Response:**
> Our motivation was to analyze methods considered before in the literature and not to invent new techniques. Given that, to the best of our knowledge, EOT was never used in the analysis of signed measures, we did not analyze it but believe that it admits similar robustness properties.  It is likely that the multiscale argument can be applied to a variant of EOT for signed measures, using a grid decomposition and controlling the entropy as is done in [1]. This will add a case distinction depending on the size of the regularization parameter, which we deem outside the scope of this first work.
>
> [1] Carlier, G., Pegon, P. & Tamanini, L. Convergence rate of general entropic optimal transport costs. Calc. Var. 62, 116 (2023). https://doi.org/10.1007/s00526-023-02455-0
>
> **Question 3:**
> Can you validate the findings on real cryo-EM data?
>
> **Response:**
> In subsection 4.3 we take projections of a real reconstruction of a molecule using the package CryoJAX. This package is aimed at accurately mimicking the forward model of the cryo-electron microscope for most numerical testing purposes.
>
> **Question 4:** Is the code available for reproducing the experiments?
>
> **Response:**
> Yes, and is attached as supplementary material. An updated version which addresses the concerns will be added with the camera-ready version.

---

> > ### Author Rebuttal · Reviewer_8uCT · 2026-04-01
> >
> > Thanks for the author's reply, the rebuttal solved my concerns.

---

### Official Review · Reviewer_EzAq · 2026-03-12

**Soundness:** 3
**Presentation:** 3
**Significance:** 1
**Originality:** 3
**Overall Recommendation:** 4
**Confidence:** 5

**Summary:**

This paper derives theoretical bounds for optimal transport–based image dissimilarity measures under a specific Gaussian noise model. The analysis considers three scenarios: (1) the deviation of a single image under perturbations, (2) the change in transport cost between two perturbed images compared to the original signed Wasserstein cost, and (3) the change in transport cost between two perturbed images relative to the original 1-Wasserstein distance. The signed Wasserstein cost is adopted to handle images with potentially non-positive intensities. Experimental results are provided to support the derived bounds.

**Compliance With Llm Reviewing Policy:**

Affirmed.

**Final Justification:**

The authors addressed my concerns for missing related work, clarified for the experimental results. Therefore I recommend weak accept.

**Key Questions For Authors:**

1. In Figure 8, the visualization does not clearly demonstrate that $W_2$ produces a better blue-to-red gradient compared to $W_1$. As a result, it is difficult to see why $W_2$ should be considered preferable in this example. Could the authors clarify how the figure supports this claim or provide a clearer visualization?

2. (line 325) The paper states that the “worst-case scenario” occurs when the images are either very similar or very far apart. Could the authors clarify why these cases correspond to the worst-case behavior? Providing additional intuition or explanation would be helpful.

3. Figure 7 is somewhat difficult to interpret from the current description. Could the authors provide additional explanation of what the figure is intended to demonstrate?

**Limitations:**

See weaknesses.

**Strengths And Weaknesses:**

Strengths:
1. The theoretical analysis is novel and clearly developed, and the experiments are closely aligned with the theoretical results, providing empirical support for the derived bounds.
2. The paper studies an important question regarding the robustness of image dissimilarity measures under noise.

Weaknesses:
1. A major concern/limitation is that the Wasserstein metric (even with signed extensions) is generally not considered a reliable measure for comparing images directly at the pixel-grid level. Two images that are close under the Wasserstein distance can still be semantically very different. While such metrics may be suitable for cryo-EM images, they are less appropriate for general natural images, where semantic similarity plays a central role. It may therefore be more appropriate to focus the scope of the paper on application domains where pixel-level transport distances are meaningful, and adjust the title accordingly to reflect this setting.

2. In the related work discussion for extensions of Wasserstein distances (line 66, right column), Transport $L^p$ distances [1] were proposed as an extension to deal with non-positive signals and should be cited.

3. The analysis is restricted to a specific Gaussian noise model, which limits the scope of the results.

4. The paper repeatedly refers to “signed Wasserstein distances” or “signed Wasserstein metrics.” For example, in lines 193, 334, etc. However, as discussed in [2], and as briefly acknowledged by the authors, this formulation defines a proper metric only when p=1. Correcting these terminologies would make the paper more rigorous.

[1] Thorpe, M., Park, S., Kolouri, S., Rohde, G.K. and Slepčev, D., 2017. A transportation L p distance for signal analysis. Journal of mathematical imaging and vision, 59(2), pp.187-210.

[2] Mainini, E., 2012. A description of transport cost for signed measures. Journal of Mathematical Sciences, 181(6), pp.837-855.

Typos:
1. (line 80) In the dual formulation, $f^d$ should be $\phi^d$

---

> ### Author Rebuttal · Authors · 2026-03-31
>
> Thank you for the careful reading and constructive feedback. A point-by-point response follows:
>
> **Weakness 1:** A major concern/limitation is that the Wasserstein metric (even with signed extensions) is generally not considered a reliable measure for comparing images directly at the pixel-grid level. [...]
>
> **Response:** In several important imaging modalities pixel brightness corresponds directly to a density of the observed object. Specifically in the high-noise regime of cryo-EM (and cryo-ET) pixel values correspond to the (tomographically-projected) density of the electric potential. In this field, the use of OT distance at the pixel level has been growing in popularity as evidenced by several recent works that we cite from 2020-2025. This is what sparked our interest in this project. However, it is not the only relevant domain which is why we think a title change would be inappropriate. For example, in medical imaging pixel values correspond to beam attenuation (e.g., bone density in CT scans) and thus optimal transport at the pixel-grid level is a reasonable choice. For example, see this paper on non-rigid registration for brain imaging:
>
> Rehman, Tauseef Ur, Eldad Haber, Gallagher Pryor, John Melonakos, and Allen Tannenbaum. “3D Nonrigid Registration via Optimal Mass Transport on the GPU.” Medical Image Analysis,  13(6), 931–940 (2009).
>
> The adoption of pixel-grid level OT has been limited in the past due to computational bottlenecks. However, this is quickly changing with improvements in computational methods, fast approximations,  and faster chips. Thus we believe that a study of the statistical properties of these methods is timely and important.
>
> **Weakness 2:** [...] Transport $L^p$ distances [1] were proposed as an extension to deal with non-positive signals and should be cited.
>
> **Response:** Thank you for drawing our attention to this paper. We will include a citation and a brief mention of it.
>
> **Weakness 3:** The analysis is restricted to a specific Gaussian noise model, which limits the scope of the results.
>
> **Response:** Our proofs can be easily extended to other noise models such as mean-corrected independent subgaussian or bounded noise. We made the classic choice to focus on Gaussian noise so that the results are not obscured by technical details. We will add a remark on this matter.
>
> **Weakness 4:** The paper repeatedly refers to “signed Wasserstein distances” or “signed Wasserstein metrics.” [...] this formulation defines a proper metric only when p=1.
>
> **Response:**
> Thank you for pointing this out. It will be corrected in the camera-ready version.
>
> **Question 1:** In Figure 8, the visualization does not clearly demonstrate that $W_2$ produces a better blue-to-red gradient compared to $W_1$.
>
> **Response:** Indeed, Figure 8 does not convincingly show this. We ran a new experiment where we compared a clean reference image to two variants: one spatially shifted (a structural change) and one corrupted by pure Gaussian noise. The result is that $W_1$ is quickly overwhelmed by the noise. $W_2$ OTOH heavily penalizes the spatial shift while absorbing the local noise. Our simulation shows $W_2$ tolerates 3.5x more noise than $W_1$ before it starts confusing additive noise with a genuine spatial shift. We believe that this new figure is a better demonstration of the improved robustness of $W_2$ in a high-noise regime.
>
> **Question 2:** (line 325) [...] Could the authors clarify why these cases correspond to the worst-case behavior?
>
> **Response:** We meant the case where the clean pictures have many zero pixels and where the signal is located in areas that are far apart. In these cases the mass has to move through large regions without signal. This extreme situation is the one for which noise corruption has the greater effect. It is then possible to shuffle noise around at low cost when matching, a somewhat expected but important phenomenon. Calling that worst-case made sense to us as this is a behavior of OT that we don’t quite want. The other extreme is when both clean pictures coincide.
>
> **Question 3:** Figure 7 is somewhat difficult to interpret from the current description.
>
> **Response:** The figure visually explains the decreasing distance phenomenon. The top row (forward transport) and bottom row (backward transport) demonstrate that as noise increases, mass is transported across a smaller distance. This shows that higher noise localizes the optimal transport plan, which lowers the Wasserstein distance. We will update the caption of Figure 7 to clarify.
>
> ===
>
> In light of our responses and the overall good scores we ask the referee if they would be willing to consider upgrading their overall recommendation. While we do not agree that the significance of our work is poor, even if that is the case, to the best of our understanding reject recommendations are reserved for cases where the paper has some critical flaw. Not for otherwise good papers with a limited scope/significance.

---

> > ### Author Rebuttal · Reviewer_EzAq · 2026-04-03
> >
> > I appreciate the authors' detailed response. I agree with the authors that this method can be useful in medical imaging. Yet I still think this title could be misleading to the broader ML community. I also hope the authors make sure to add the new figure they mentioned to better illustrate the benefit of using $W_2$ in the revision. I will raise my score.

---

### Decision · Program_Chairs · 2026-04-30

**Decision:**

Accept (regular)

**Comment:**

The paper presents theoretical bounds for optimal transport–based image dissimilarity measures under a specific Gaussian noise model. Here images are modeled as signed measures supported on a grid and the signed Wasserstein distance is considered. In particular, the authors provide insight into why the signed Wasserstein distance can be advantageous in high noise setting, as it is more robust to noise compared to the Euclidean norm. Numerical experiments are provided to illustrate the theoretical bounds.

Reviewers are generally positive about the theoretical analysis presented in the paper, especially concerning the signed Wasserstein distance, which has not received much attention in the machine learning literature. Concerns from reviewers include: the signed Wasserstein distance is truly a distance (i.e. satisfies the triangle inequality) only when p = 1; the Wasserstein distance may be suitable for cryo-EM images but not for natural images (Reviewer  EzAq); the bounds are not tight (Reviewer 8uCT, also acknowledged by the authors themselves)

The scores are Weak Accept, Accept, Weak Accept, Weak Accept.